

# Sensitivities of atmospheric composition and climate to altitude and latitude of hypersonic aircraft emissions

Johannes Pletzer[1,2], Volker Grewe[1,2]

[1]Deutsches Zentrum für Luft- und Raumfahrt, Institut für Physik der Atmosphäre, Oberpfaffenhofen, Germany
[2]Delft University of Technology, Aircraft Noise and Climate Effects, Delft, The Netherlands

*Correspondence to*: Johannes Pletzer (johannes.pletzer@posteo.com)

**Abstract.** Hydrogen-powered hypersonic aircraft are designed to travel in the middle stratosphere at approximately 30-40 km. These aircraft can have a considerable impact on climate-relevant species like stratospheric water vapour, ozone and

methane and thus would contribute to climate warming. The impact of hypersonic aircraft emissions on atmospheric composition and in turn on radiation fluxes differs strongly depending on cruise altitude. However, in contrast to variation of altitude of emission, differences from variation of latitude of emission are currently unknown. Here we show that a variation in latitude of emission can have a larger effect on perturbations and stratospheric-adjusted radiative forcing than a variation of altitude of emission. Our results include individual impacts on middle atmospheric water vapour, ozone and methane, of

water vapour and nitrogen oxide emissions, as well as unburnt hydrogen and the resulting radiative forcing. Water vapour perturbation lifetime continues the known tropospheric increase with altitude and reaches almost six years in the middle stratosphere. Our results demonstrate how atmospheric composition changes caused by emissions of hypersonic aircraft are controlled by large-scale processes like the Brewer-Dobson circulation and, depending on latitude of emission, local phenomena like polar stratospheric clouds.

The analysis includes a model evaluation of ozone and water vapour with satellite data and a novel approach to reduce simulated years by one-third and thus cost and climate impact. A prospect is the analysis of seasonal sensitivities and simulations with emissions from combustion of liquefied natural gas instead of liquid hydrogen.

## 1. Introduction

Since the civil supersonic aircraft Concorde and Tupolev Tu-144 entered and left service (1969-2003), some time has

passed. In the meantime new projects appeared that intend to solve the predecessors problems, which were mainly the large cost, the significant contribution to climate warming and the strong noise. More than a few manufactures, e.g. *Boom*



*Technology*[1] or *Spike Aerospace*[2], are on their way to deliver supersonic aircraft for commercial use. But, to our knowledge, a full-scale prototype has not been announced until today by any of the manufacturers.

The supersonic aircraft design with the best market potential is a small scale business jet for customers that value time greatly (Sun and Smith, 2017; Liebhardt et al., 2011). Hence, the biggest benefit of and best sale argument for supersonic business jets are their speed and their travel time. For the same distance, the travel time of supersonic aircraft is advertised as approximately half that of subsonic aircraft. Of course the different supersonic aircraft designs each have their specific speed and may deviate from this number. In comparison, hypersonic aircraft, which are in development, but not as technologically advanced as supersonic aircraft, fly at higher altitudes and higher speed compared to supersonic aircraft. Hence, they reduce travel time by a factor four to eight instead of two.

Clearly, on one hand, the aircraft industry strives to satisfy customer's needs with potential innovations like super- and hypersonic aircraft. On the other hand, however, aircraft industry currently is in a transformation process to build a climate compatible aircraft infrastructure. It easily takes decades to develop new aircraft before they can be launched on the market and the lifetime of aircraft can also be counted in decades. Hence, the climate compatibility of new aircraft has a long-lasting effect and is therefore of utmost importance.

Generally, numerous estimates of the climate impact and the growth potential of the current aircraft industry have been published (Lee et al., 2021; Fahey et al., 2016; Pachauri, R.K. and Reisinger, A. (Eds.), 2007; Pachauri, R.K and Meyer, L.A. (Eds.), 2014). One of the newest analyses the climate impact and the growth potential using projections of different technological development scenarios (Grewe et al., 2021).

The impact on atmosphere and climate of potential fleets of supersonic aircraft that could replace parts of a conventional fleet, has been estimated and reviewed in many - some very extensive - publications (Grewe et al., 2010; Zhang et al., 2021a; Penner, J.E. et al., 1999; Matthes et al., 2022). In contrast, first publications about atmospheric perturbation and climate impact of hypersonic aircraft have only appeared more recently (Ingenito, 2018; Kinnison et al., 2020a; Pletzer et al., 2022).

Briefly summarized for supersonic and hypersonic aircraft, the non-$CO_2$ effect, mainly water vapour ($H_2O$), contributes more to climate impact with increasing altitude due to the longer $H_2O$ perturbation lifetime and the resulting radiative forcing and climate change is manifold that of subsonic aircraft (Grewe et al., 2010; Pletzer et al., 2022). The depletion of the ozone ($O_3$) layer is significant, however, the altitude dependency of the radiative forcing to emission altitude is more complex for $O_3$ than for $H_2O$ (Zhang et al., 2021b, c).

---

[1] https://boomsupersonic.com/company

[2] https://www.spikeaerospace.com/





Since the development of high-flying aircraft is clearly continuing, the application of mitigation options for the lowest possible climate impact during commercial use are crucial and should be fully exploited. That is not only in case these aircraft really are in use one day, but also to estimate the optimization potential beforehand. The technological reality that any fuel will ultimately increase the concentration of the main climate driver $H_2O$ seems inevitable, since purely battery-powered aircraft are not a viable option for distances that super- and hypersonic aircraft can cover in a single flight due to the batteries weight. Taking 100 % sustainable alternative fuels provides the option of net-zero $CO_2$ emission flights, although availability and the degree of sustainability is likely to be a limitation for the coming decades, and non-$CO_2$ effects remain. Therefore, the importance of improving in-flight climate efficiency of fueled aircraft cannot be emphasized enough. The potential of climate optimized routing for conventional aircraft (Grewe et al., 2014a; Grewe et al., 2014b; Sridhar et al., 2011; Matthes et al., 2012, 2020) and climate optimized design (Grewe et al., 2017; Dahlmann et al., 2016) has been shown before. To achieve the same for high-flying aircraft, it is key to analyze sensitivities at stratospheric altitudes i.e. the impact of aircraft emissions on atmospheric composition and climate depending on latitude and altitude of emission.

Therefore, in this study, we performed 24 simulations and a reference simulation with the Atmospheric-Chemistry General Circulation Model ECHAM/MESSy (EMAC)j, addressing a wide range of sensitivities. The simulations include emission scenarios where $H_2O$, $NO_x$ and $H_2$ are emitted separately at two altitudes (30 km, 38 km) and four latitude regions (southern mid-latitudes, tropics, northern mid-latitudes, north polar). The atmospheric composition changes are then used to calculate the radiative forcing with EMAC.

The paper is structured as follows. In section three, we present the EMAC model and the simulation setup including a model evaluation with $H_2O$ and $O_3$ satellite measurements and a novel approach to reduce simulation cost. Section four addresses the variation of emission in latitude and altitude and the emissions' magnitude. The direct and indirect effects of emissions, $H_2O$, $NO_x$ and $H_2$, on atmospheric composition is shown in section five. Section six focuses on the radiative forcing from the atmospheric composition changes, followed by a discussion of atmospheric and radiative sensitivities in section seven, a general discussion in section eight and a summary.

## 2. Methods and simulations

### 2.1. EMAC

We used ECHAM5/MESSy to calculate atmospheric composition changes and radiative forcing. ECHAM5 is the dynamical core of EMAC and used to calculate the atmospheric general circulation (Roeckner et al., 2006). MESSy, version 2.54.0 of the Modular Earth Model System (Jöckel et al., 2016a, 2005, 2006; Jöckel et al., 2010), serves as a connector with its modular infrastructure and 'plug-in' submodels. Overall, ECHAM5/MESSy is able to combine the atmosphere, ocean and



land domain and the chemistry within MESSy is interconnected with these three domains. Therefore, EMAC is a full-scale Earth System Model (ESM), if needed. However, EMAC can be used for a variety of applications. Our specific setup, which is not a full-scale ESM, but an Atmospheric Chemistry General Circulation Model (AC-GCM), is presented in the following subsection and follows the setup already presented in Pletzer et al. (2022).

## 2.2. EMAC model setup

In our simulation setup we used the following submodels for physics: AEROPT (Aerosol OPTical properties), CLOUD, CLOUDOPT (CLOUD OPTical properties), CONVECT (CONVECTion), CVTRANS (ConVective Tracer TRANSport), E5VDIFF (ECHAM5 Vertical DIFFusion), GWAVE (Gravity WAVE), OROGW (OROgraphic Gravity Wave), QBO (Quasi Biannual Oscillation) and RAD (RADiation).

AEROPT calculates aerosol optical properties and CLOUDOPT cloud optical properties (Dietmüller et al., 2016). CLOUD accounts for cloud cover, cloud micro-physics and precipitation and is based on the original ECHAM5 subroutines, as is CONVECT, which calculates convection processes (Roeckner et al., 2006). CVTRANS is directly linked and calculates the transport of tracers due to convection. E5VDIFF addresses vertical diffusion and land-atmosphere exchanges excluding tracers. GWAVE and OROGW account for (non-)orographic gravity waves. QBO includes winds of observed quasi-biannual-oscillations. RAD and RAD_FUBRAD contain the extended ECHAM5 radiation scheme and allow multiple radiation calls like stratospheric adjusted radiative forcing (Dietmüller et al., 2016).

For chemistry, we used the following submodels: AIRSEA, CH4, DDEP (Dry DEPosition), H2OEMIS (H2O EMISsion), JVAL (J Values), LNOX (Lightning Nitrogen OXides), MECCA (Module Efficiently Calculating the Chemistry of the Atmosphere), MSBM (Multiphase Stratospheric Box Model), OFFEMIS (OFFline EMISsion), ONEMIS (ONline EMISsion), SCAV (SCAVenging), SEDI (SEDImentation), SURFACE and TNUDGE (Tracer NUDG(E)ing) and TREXP (Tracer Release EXPeriments).

AIRSEA calculates the deposition and emission over the ocean, which corrects the global budget of tracers like methanol and acetone (Pozzer et al., 2006). CH4 determines the oxidation of methane ($CH_4$) by the hydroxyl radical OH, atomic oxygen $O(^1D)$, chlorine and photolysis (Winterstein and Jöckel, 2021). In our setup the feedback to specific humidity is activated in MECCA. DDEP accounts for the dry deposition of aerosol tracers and gas phase tracers and SEDI for the sedimentation of aerosols and their components (Kerkweg et al., 2006a). H2OEMIS adds $H_2O$ emissions to specific humidity via TENDENCY (Pletzer et al., 2022). JVAL calculates the photolysis rate coefficients (Sander et al., 2014). LNOX contributes a parametrization (Grewe) for $NO_x$ production by lightning for intra-clouds and clouds-to-ground (Grewe et al., 2001; Tost et al., 2007). MECCA accounts for all internal tracers related to production and destruction of chemical components (Sander et al., 2005; Sander et al., 2011). MSBM calculates the polar stratospheric cloud-chemistry and is based



on the PSC (Polar Stratospheric Cloud) submodel code (Jöckel et al., 2010). The representation of polar stratospheric clouds includes three subtypes of polar stratospheric clouds, i.e. solid nitric acid trihydrate (NAT) particles (type 1a), super-cooled

ternary solutions ($HNO_3$, $H_2SO_4$, $H_2O$, type 1b) and solid ice particles (type 2) (Kirner et al., 2011a). The latter start to form below the frost point. Multiple parameters regarding polar stratospheric clouds are written as output within EMAC. Examples are mixing ratios of relevant trace gases ($HNO_3$, HCl, HBr, HOCl, HOBr) in liquid, solid or gas phase, number densities of ice and NAT particles and loss and production of NAT through sedimentation. Further included are the physical parameters velocity and radius and surface of particles. OFFEMIS prescribes emission data like relative concentration

pathways (RCPs) and ONEMIS calculates emissions online; both are added to internal tracer tendencies (Kerkweg et al., 2006b). SCAV accounts for wet deposition and liquid phase chemistry in precipitation fluxes (Tost et al., 2006). SURFACE originates from several ECHAM5 subroutines and calculates temperatures of different surfaces (Roeckner et al., 2006). TNUDGE is responsible for tracer nudging (Kerkweg et al., 2006b).

The applied horizontal resolution is T42 (triangular truncation at wave number 42) for the dynamical core, which relates to a

Gaussian grid of approximately 2.8° x 2.8° in longitude and latitude. The vertical resolution includes 90 levels from the surface to 0.01 hPa (center of the uppermost layer) of the middle atmosphere (MA) at approximately 80 km. The complete resolution description is abbreviated as T42L90MA.

We used specified dynamics i.e. we nudged towards ERA-Interim reanalysis data. This Newtonian relaxation is applied for the GCM variables divergence with a relaxation timescale of 48 h, vorticity (6 h) and the logarithm of the surface pressure

(24 h). The domain, where nudging is applied, ranges from the 4[th] model layer above the ground to approximately 200 hPa.

Since a potential entry-into-service of hypersonic aircraft is approximately 2050, we used the CCMI RCP6.0 scenario to simulate atmospheric conditions starting 2050. Hence, these surface emissions were included as lower boundary conditions and added to the tracers. The aircraft emission scenarios were excluded.

Temperature, $O_3$ and $H_2O$ representation of the upper troposphere-lower stratosphere (UTLS) region in EMAC were

evaluated using aircraft measurements (Pletzer et al., 2022). We kindly ask the reader to refer to this publication for detailed information. To summarize briefly, the key message for EMAC is, firstly, a systematic cold bias of -3.8 to -2.5 K in the extra-tropics, responsible for an upward shift of the tropopause and as a consequence an under- and overestimation of $O_3$ and $H_2O$ mixing ratios, respectively. Secondly, the Taylor correlation coefficient, estimating correlation between observations and model results in the UTLS, is $r \sim 0.90$ for $H_2O$ and $r \sim 0.95$ for $O_3$ and temperature. In this publication, we extend the

verification region to higher altitudes with satellite measurements published as the 'Stratospheric Water and OzOne Satellite Homogenized data set' SWOOSH by Davis et al. (2016). Further information are presented in the following subsection Sec. 3.3.



### 2.3. Evaluation with satellite data (SWOOSH)

The satellite dataset SWOOSH contains $H_2O$ and $O_3$ mixing ratios for the time period 1984 to 2022 and data is available for
nearly all the stratosphere and parts of the mesosphere. Here, we compare a multi-annual mean of measurements for the
years 2013-2016 to two kinds of modeled stratospheric $O_3$ and $H_2O$. First, to the EMAC simulations RC1SD-base-10 from
Jöckel et al. (2016b), with present day meteorology and, second, to our model simulation results. Note that in contrast to the
former setup, we are using boundary conditions (RCP6.0) for the year 2050. Hence, $CH_4$ sources, nitrous oxide, chlorine and
bromine source gases differ between the simulations, which in turn causes differences in stratospheric $H_2O$ and $O_3$
concentrations. While the model setups of RC1SD-base-10 and ours are largely the same, we address the differences due to
the 2050 boundary conditions in our model for $O_3$ and $H_2O$.

### 2.3.1. SWOOSH introduction and evaluation

The SWOOSH data consists of $O_3$ and $H_2O$ measured with the satellite Aura MLS and shows no gaps for the years 2013-
2016. To compare, Aura MLS measurements of $H_2O$ are quite close to the multi-instrumental mean (MIM), published by the
SPARC Data Initiative (Fig. 14, Hegglin et al., 2013, p.11), with a continuous and moderate overestimation at all pressure
levels and for both, tropics and extra-tropics. Aura MLS measurements of $O_3$ agree very well with the MIM at 100-5 hPa and
show a small underestimation at 5-1 hPa for both, tropics and mid-latitudes (Fig. 10, Tegtmeier et al., 2013, p.12).

We used version 2.6 of the SWOOSH data with a horizontal resolution of 2.5° and 31 vertical levels. The vertical limits are
316-0.002 hPa for $H_2O$ and 261-0.02 hPa for $O_3$. The vertical resolution of SWOOSH data is 2.5-3.5 km for both, $O_3$ and
$H_2O$. For the comparison, EMAC data was interpolated to vertical and horizontal levels of SWOOSH data.

### 2.3.2. EMAC water vapour evaluation

Fig. 11 shows the feature of increasing $H_2O$ mixing ratios with altitude for both, model results and observations. Clearly,
observation features are not as homogeneous as model results, but the general agreement is very good. The relative
differences show that the magnitude of $H_2O$ mixing ratios in the model setup with present day meteorology is 46±13 % and
our EMAC model setup is 33±8 % smaller than observations on average. The largest differences appear particularly at
tropical and southern latitudes in the lower stratosphere.

The increase in average $CH_4$ mixing ratio of the RCP scenario between the 2010s and 2050s is about 6 %, which increases
stratospheric $H_2O$ and could explain the larger difference of observations to RC1SD-base-10 simulations compared to our
model setup. Note that differences between model and observations should in reality be less than shown in Fig. 11, since
$H_2O$ of Aura MLS is slightly overestimated at these altitudes, compared to the MIM of the SPARC Data Initiative. In
summary, we conclude that EMAC underestimates the $H_2O$ mixing ratio at stratospheric altitudes for the RC1SD-base-10





results, which are based on trace gas measurements from 2013-2016. For our model setup results, which are a projection to 2050, we expect the difference to RC1SD-base-10 to originate mostly from the difference in lower boundary conditions of $CH_4$.

### 175 **2.3.3. EMAC ozone evaluation**

The multi-annual mean of $O_3$ shows very good agreement between model and observations (Fig. 12). Clearly, the features look the same and the difference in average magnitude between observation and model is rather small with -5±6 % and -12±6 % for RC1SD-base-10 (present day meteorology) and for our model setup (projection to 2050), respectively. For the RC1SD-base-10 scenario SWOOSH observations are within the standard deviation of model simulations. The largest
deviations appear in the same areas for both model setups, RC1SD-base-10 and ours. They are at the tropics and southern mid-latitudes at 100 hPa and at polar regions at higher altitudes.

$O_3$ mixing ratios are expected to increase over time compared to the 2010s, since emission of chlorine and bromine source gases stagnate significantly (World Meteorological Organization et al., 2019; Organization (WMO) et al., 2015). Hence, the difference, when comparing measurements of the 2010s and model results with boundary conditions for the 2050s, is to be
expected. Therefore, we conclude that EMAC shows a very good agreement of stratospheric $O_3$ mixing ratios compared to observations for both our model results and the RC1SD-base-10 scenario.

### **2.4. Enhancing the efficient use of computing resources**

The simulations with Earth System Models are very time and cost intensive with a high energy consumption. Therefore, we introduce a method, which reduces the number of simulated years of the spin-up phase to three-fifths.

Finite computer resources were mentioned before by Hasselmann et al. (1993) and the related 'cold start' error. The 'cold start' problem refers to coupled atmosphere-ocean simulations that are started close to the present and not early at the beginning of anthropogenic greenhouse gas emission, which eventually generates a significant error in the forcing results due to the too short spin-up phase.

Our computing challenge has an analogy to the above described coupled Atmosphere-Ocean GCM, that are normally run for
very long time scales and, therefore have a large reduction potential. Here, we run an Atmospheric-Chemistry GCM for many simulations with variations in the location of emitted trace gases at stratospheric altitudes, each requiring a significant spin-up time (Fig. 3, Pletzer et al., 2022).

With our applied speed-up method, we achieve equilibrium on the multi-annual mean faster and hence are able to reduce the simulated years to two-thirds. The method is the following. We estimate the equilibrium mass change of a species due to an



emission and apply an increase of the emissions with a factor $s$ in the first year, so that this equilibrium mass change is approximately reached after a one-year simulation, instead of a 10 year spin-up. The factor $s$ is calculated using a one-dimensional differential equation containing the $H_2O$ perturbation lifetime $\tau$ and is therefore depending on altitude (Fig. 13). In a quasi-steady state with a stratospheric $H_2O$ mass perturbation $X$ (kg), an annual emission $E$ (kg/year) and a perturbation lifetime $\tau$ (years) we obtain

$$0 = \frac{\delta X}{\delta t} = E - \frac{1}{\tau}X, \qquad (1)$$

or in other words

$$X = \tau E. \qquad (2)$$

To obtain the value of $s$ we solve the differential equation for the evolution of the stratospheric $H_2O$ mass change to $Y$ for an emission $sX$:

$$\frac{\delta Y}{\delta t} = sE - \frac{1}{\tau}Y, \qquad (3)$$

with the boundary conditions $Y(t = 0) = 0$ and $Y(t = 1) = \tau E$. The solution of the differential equation, that fulfills the first boundary condition is then

$$Y(t) = s\tau E \left(1 - e^{-\frac{1}{\tau}t}\right) \qquad (4)$$

In combination with the second boundary condition this gives us

$$s = \frac{1}{1 - e^{-\frac{1}{\tau}}} \qquad (5)$$

Hence, if we estimate $s$ with a Taylor approximation we obtain the perturbation lifetime of $\tau$. However, if we enhance the emission by this factor, we additionally have to consider the loss during the first year with a factor of 0.5, which applies for $\tau$ between 2 to 5 years.

$$s \approx \tau + 0.5 \qquad (6)$$

We used $H_2O$ perturbation lifetime values at stratospheric altitudes from Grewe and Stenke (2008) and Pletzer et al. (2022) as initial values and inter- and extrapolated to altitudes of 30 km and 38 km, respectively. Since the slope of linear and





quadratic extrapolation (35-42 km) were either very steep or too curved we used the average of both values. Using the interpolated $\tau$ values we calculated $s$ using Eq. (6).

The process during the model runs is as follows. After the first year of simulation, where we enhanced the emissions by the factor $s$, the initial values of trace gases are emitted for the following years in the simulation. The first year, on one hand, ensures that atmospheric composition changes reach multi-annual equilibrium faster. On the other hand, by limiting the enhancement to the first year, the final mass perturbation is not disturbed, since the atmospheric lifetime of perturbations is shorter than the spin-up phase. For validation, we compared the mass perturbation of $H_2O$ above the tropopause over time for the different scenarios, which is shown in Fig. 14. Please refer to subsection Sec. 4.3 for a comparison of scenarios and

validation of multi-annual mean equilibrium. To summarize briefly, with the altered spin-up we reduced consumption of normally used computational resources to two-thirds with minimal error potential for all 24 perturbation simulations.

## 3. Emission scenarios

We use total annual emissions of a potential fleet of hypersonic aircraft that consist of $H_2O$, $NO_x$ and $H_2$ emissions. These emissions were presented before by Pletzer et al. (2022) { page 12 } and originate from the 'HIgh speed Key technologies

for future Air transport - Research & Innovation cooperation scheme'-project, abbreviated as HIKARI (Blanvillain and Gallic, 2015). $H_2O$ emissions occur as a product of $H_2$ combustion. Nitrogen oxide is produced at high temperatures during combustion and originate from ambient air ($N_2$ and $O_2$). Complete combustion of $H_2$ is very difficult to achieve and in this dataset it is estimated that 10% of the $H_2$ fuel remains unburned. These values exclude tropospheric and subsonic emissions and are from altitudes above 18 km, referring to exhaust during cruise phase. The following two subsections contain

information on the location and magnitude of all 24 perturbation scenarios.

### 3.1. Emission locations of perturbation scenarios

The emission scenarios are not based on aircraft route networks, but are box emissions that are varied in latitude and altitude. Altitudes of emission are 30 km and 38 km. Note that the height of the box, where the exhaust is emitted, is approximately 1 km. Therefore, it is rather an altitude range with 30-31 km and 38-39 km, where trace gases are emitted. Each latitude region

of emission spans 30°. They are southern mid-latitudes at 60-30° S, northern tropics at 0-30° N, northern mid-latitudes at 30-60° N and north polar latitudes at 60-90° N. Clearly, south polar and southern tropics are missing. We refrained from including these two regions for three reasons. First, aircraft barely fly at southern polar regions, since most direct routes between city-pairs do not cross there. Therefore, they are not as important as an emission location. Second, since we included northern tropics, we passed on southern tropics since they show very similar dynamic and chemical conditions in





the stratosphere. However, we included southern mid-latitudes to quantify the impact on atmospheric composition and climate on the Southern Hemisphere. Third, with this approach we could reduce the use of computational resources further.

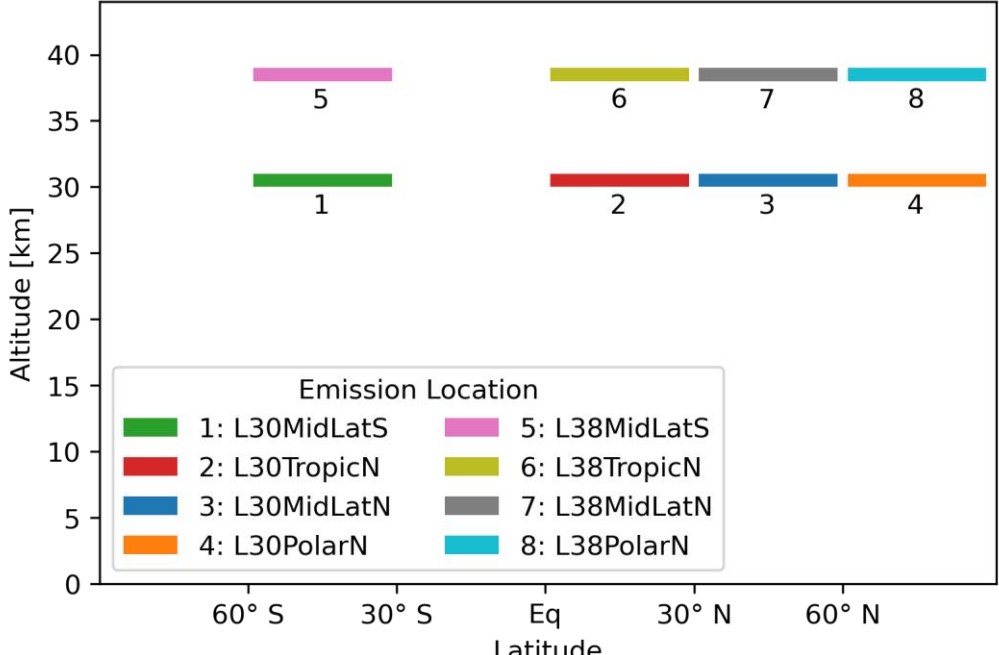

**Figure 1: Altitude and latitude region of emission of in total eight emission locations. For three types of emission these sum to 24 emission scenarios.**

**3.2. Emission magnitude of perturbation scenarios**

During simulations, the individual trace gases (Tbl. 1) are continuously emitted in the described altitude and latitude regions and with that introduced to physical and chemical processes in the AC-GCM. We aim at identifying the impact on the atmosphere from the cruise phase and hence do not consider other phases such as take-off and landing in this idealized simulation setup.

For each emitted trace gas ($H_2O$, $NO_x$, $H_2$) we have a total of eight simulations, which sum up to 24 simulations in total. The annual magnitude of emitted trace gases is 21.24 teragram of $H_2O$, 0.031 teragram $NO_2$ of $NO_x$ and 0.236 teragram of $H_2$. These values are based on emissions of the aircraft design LAPCAT-PREPHA (Long-Term Advanced Propulsion Concepts and Technologies; Programme de REcherche et de technologie sur la Propulsion Hypersonique Avancée).

**Table 1: Emission scenarios listed by the kind of emission and magnitude. Information on the total number of simulations, including meridional and altitude distribution, are given in two additional columns.**

| Emission | Magnitude | Magnitude | Number of locations | Number of altitudes | Number of simulations |
|---|---|---|---|---|---|





| Emission | Magnitude | Magnitude | Number of locations | Number of altitudes | Number of simulations |
|---|---|---|---|---|---|
| $H_2O$ | 21.24 Tg/year | 1.18 Tmol | 4 | 2 | 8 |
| $NO_x$ | 0.031 $TgNO_2$/year | 0.67 Gmol | 4 | 2 | 8 |
| $H_2$ | 0.236 Tg/year | 0.118 Tmol | 4 | 2 | 8 |

### 3.3. Timeline and validation

To validate the multi-annual mean equilibrium of atmospheric composition in the context of the speed-up, presented in subsection Sec. 3.4, we compare the timeline of $H_2O$ mass perturbations. Of the emitted trace gases $H_2O$ has the longest perturbation lifetime and therefore serves as the medium to verify multi-annual mean equilibrium. Fig. 14 shows the
temporal evolution of $H_2O$ mass perturbation. Three phases can be seen. (1) The first year shows a sharp increase due to the applied speed-up factor $s$ (gray shaded area). (2) From 2008-2012, some scenarios overshot equilibrium values with the end of the first year, e.g. 30 km, 60-90°N, and level off to equilibrium values. Other scenarios continue to build up, e.g. most scenarios at 38 km, before reaching equilibrium (green shaded) latest in 2013 (3).

The different behavior most probably originates from latitudinal differences in $H_2O$ perturbation lifetime, which were not
included in the inter- and extrapolation in subsection Sec. 3.4. The error potential should not be significant, since the time to equilibrate is sufficient for all scenarios and multi-annual mean equilibrium is reached latest in 2013. Clearly, equilibrium is not reached within one year as assumed and total spin-up until equilibrium amounts to 6 years, since latitudinal differences had to be taken into account as well. However, without the spin-up method (Sec. 3.4), we would require 10 years (Fig. 3, Pletzer et al., 2022) and hence we could reduce the spin-up computing time by 40 % and total computing time by
approximately 33 %.

### 4. Atmospheric composition changes

In this section we present, first, as an overview of simulation results, i.e. the impact on the two most important climate active gases in the context of hypersonic aircraft exhaust, $O_3$ and $H_2O$, and, second, direct and indirect effects on atmospheric composition by the emission of $H_2O$, $NO_x$ and $H_2$ for all 24 scenarios. Third, we address $CH_4$ and $H_2O$ perturbation lifetime.
We want to mention that all presented data in this section is based on a multi-annual mean for the years 2013-2016.





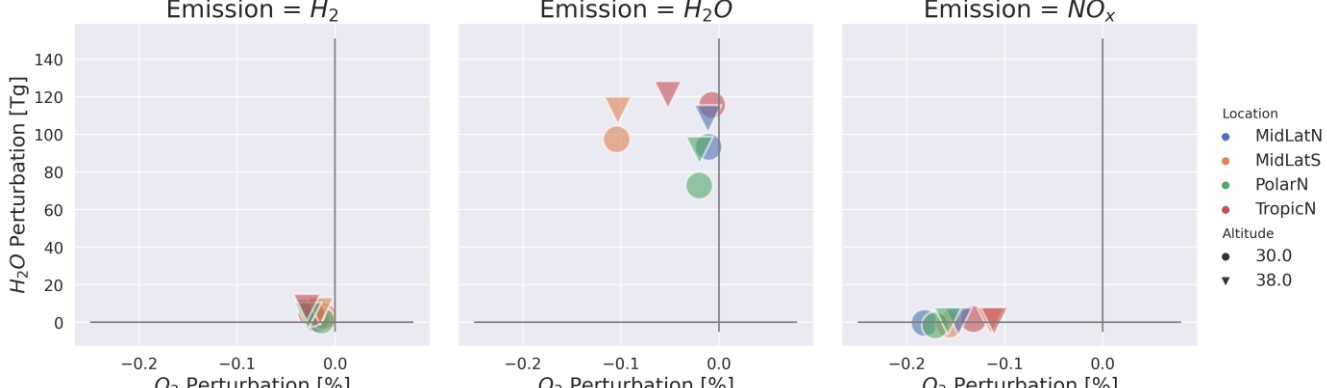

**Figure 2: Overview of H₂O and O₃ changes for all 24 simulations grouped by initial emission of H₂, H₂O and NOₓ(left, middle, right). Colours refer to the latitude region of emission (southern mid-latitudes, northern tropical latitudes, northern mid-latitudes, north polar latitudes) and markers to altitude of emission. Please refer to Table 3 for statistical significance.**

Fig. 2 shows the H₂O mass perturbation and relative O₃ changes above the meteorological tropopause (WMO) grouped by emitted species. Hydrogen (H₂) emission has a comparably small effect on both, H₂O and O₃. Relative O₃ changes range from -0.01 % to -0.03 %, with a decrease for all H₂ emission scenarios. The effect is larger for the higher altitude scenarios. H₂O changes are not larger than 8 Tg, with maximum values for the northern tropic emission scenario at 38 km. H₂O emission has a large effect on H₂O mass perturbation in all scenarios with a range of 73-121 Tg. Values for higher altitude and lower latitude scenarios are particularly large. Relative O₃ change is rather small, except in the southern mid-latitudes where O₃ depletion is approximately 0.1% for both altitudes. The altitude of emission has a large effect for emission at the tropics and emission at 38 km has a significantly larger impact on ozone. NOₓ emission has the lowest effect on H₂O mass perturbation. The range of H₂O changes is -1.6 to 1.7 Tg and all higher altitude emission scenarios show an increase with medium values. In contrast, the lower altitude emission scenarios show larger values, but differ in increase and decrease. Relative O₃ change due to NOₓ emission is larger than most other emission scenarios with a range of -0.11 to -0.18 %. Largest values appear at northern mid-latitudes and polar regions at 30 km.

The following subsections are further divided in 'Direct Effects' and 'Indirect Effects'. The former relates to the emission and its effect on the same molecular atmospheric compound, e.g. H₂O emission on H₂O trace gas concentrations. The latter includes indirect effects on other atmospheric compounds like CH₄ or O₃. The shaded box in each subfigure represents the altitude of emission. Hatched areas represent a p-value larger than written in the title of each subplot (Tbl. 2).

**Table 2: Markers for probability levels of significance test**

| Probability | $p \gtrless 0.05$ | $p \gtrless 0.01$ | $p \gtrless 0.001$ |
|---|---|---|---|
| Marker | * | ** | *** |




### 4.1. Water vapour emission

#### 4.1.1. Direct effects

Fig. 3 shows the $H_2O$ perturbation in parts per million volume for the $H_2O$ emission scenarios. When comparing the altitudes 38 km and 30 km, which are the first and second line, respectively, the $H_2O$ mixing ratios are generally higher for the former. $H_2O$ emissions at low latitudes are distributed across both hemispheres. $H_2O$ emissions at higher altitudes are to a significantly larger extent confined to one hemisphere. The threshold for failing the t-test is exceeded at and below the tropopause region.

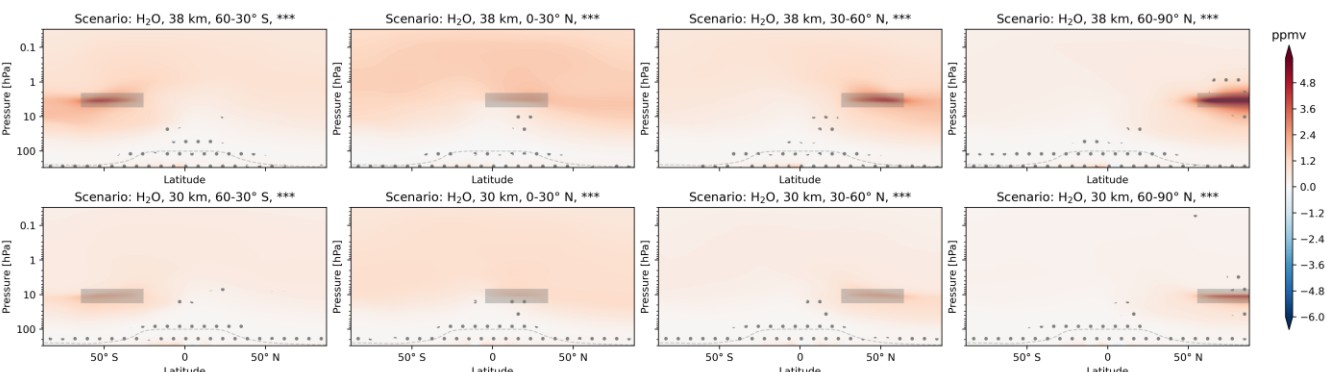


**Figure 3: Zonal mean of $H_2O$ perturbation (volume mixing ratio, ppmv) for scenarios where $H_2O$ is emitted. The first and second row refer to 38 km and 30 km altitude of emission, respectively. Columns represent the latitude region of emission. The emission regions are shaded in grey. Dots indicate statistical insignificant results and the probability level is written in the title of each subplot.**

#### 4.1.2. Indirect effects

##### *4.1.2.1. Ozone*

In total, the $H_2O$ emission causes $O_3$ depletion, especially in the southern latitudes (Fig. 2). In Fig. 4 the $O_3$ perturbation (volume mixing ratio) is divided into layers of $O_3$ decrease (above 3 hPa and below 10 hPa) and $O_3$ increase (3-10 hPa) in all subfigures. For emission at 38 km, the upper layer of $O_3$ decrease seems to be strengthened. For emission at 30 km, the layer

of $O_3$ increase overlaps with the emission region and seems to be weakened, especially for $H_2O$ emission from 0-30° N. The threshold for failing the t-test has in general a large variation and appears at the interface of $O_3$ increase and $O_3$ decrease regions, the tropopause and polar regions. The largest values of $O_3$ increase is situated in the tropics for all scenarios. In contrast, the largest values of $O_3$ decrease move with the emission location.





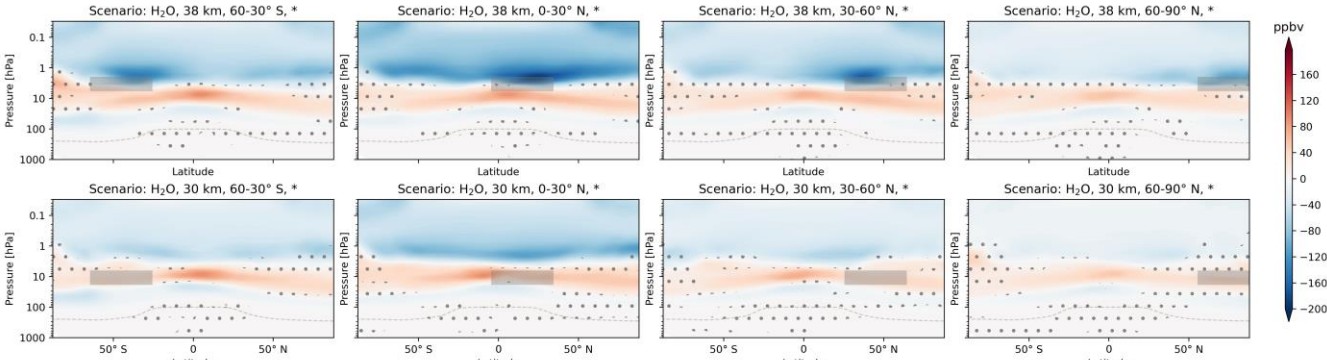

**Figure 4: Zonal mean of O₃ perturbation (volume mixing ratio, ppbv) for scenarios where H₂O is emitted. The first and second row refer to 38 km and 30 km altitude of emission, respectively. Columns represent the latitude region of emission. The emission regions are shaded in grey. Dots indicate statistical insignificant results and the probability level is written in the title of each subplot.**

*4.1.2.2. Hydrogen*

The $H_2O$ emission has a significant effect on $H_2$ (Fig. 17) at the top of the modelled atmosphere (around 80 km). In general, the $H_2$ perturbation patterns are very similar with a broad layer at high latitudes and smaller width at mid to low latitudes. The magnitude, however, is larger for emission scenarios with low latitudes and the higher emission altitude (38 km).

*4.1.2.3. Methane*

The impact of $H_2O$ emission on $CH_4$ shows complex patterns, with areas of increase and decrease, and the multi-annual variability is clearly large, since the t-test restricts confidence in many areas (Fig. 5). It is common knowledge, that OH oxidizes $CH_4$ and adds to $H_2O$ concentrations in the stratosphere. Additionally, it was shown, that emission of $H_2O$ eventually increase $CH_4$ oxidation (Pletzer et al., 2022). Overall, $H_2O$ emissions reduce $CH_4$ concentrations the most compared to emissions of $NO_x$ and $H_2$ (that changes when normalized to the number of emitted molecules). The features for emission at southern mid-latitudes are very similar with a decrease from 100-10 hPa and an additional decrease around the emission location of the higher altitude scenario. $H_2O$ emission at northern tropics causes a wide-spread $CH_4$ depletion with a larger impact for the higher altitude emission. A $CH_4$ increase is visible in the tropics at 10 hPa for 30 km emission. For 38 km, the area with $CH_4$ increase, where $p \gtrsim 0.05$, is barely visible. Northern mid-latitude emission of $H_2O$ shows features similar to southern mid-latitude emission, but the decrease is at slightly higher altitudes. Areas of increase and decrease seem to have switched location. This statement should be taken with caution, since not all areas, included in the comparison, fulfil the t-test criteria, i.e. the error probability that means are not different is larger than 5 %. North polar emission scenarios are very similar to the north mid-latitude emission scenarios. However, perturbations of volume mixing ratios are generally smaller.





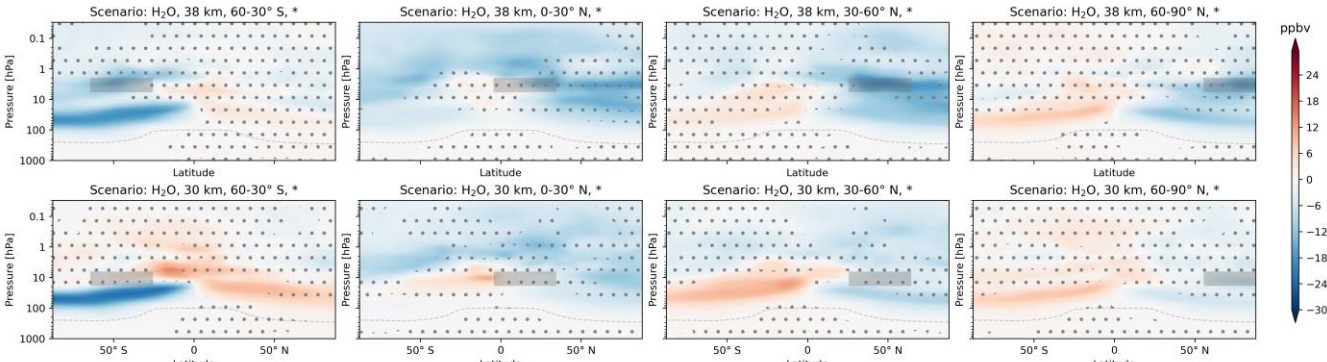

**Figure 5: Zonal mean of CH₄ perturbation (volume mixing ratio, ppbv) for scenarios where H₂O is emitted. The first and second row refer to 38 km and 30 km altitude of emission, respectively. Columns represent the latitude region of emission. The emission regions are shaded in grey. Dots indicate statistical insignificant results and the probability level is written in the title of each subplot.**

## 4.2. Nitrogen oxide emission and ozone perturbation

### 4.2.1. Direct effects

The emitted $NO_x$ is, compared to $H_2O$, more confined to the emission region. First, the $NO_x$ perturbation maximum is clearly located at the altitude and latitude of emission — other maxima are not visible. Second, the multi-annual mean of emission scenarios and the reference scenario are different for most areas in direct proximity to the emission location (hatched area). $NO_x$ changes are more distributed for low latitude emission scenarios. The vertical distribution shows downward transport for high latitudes and upward transport for low latitudes, depending on latitude of emission. A correlation plot of significant $NO_x$ and $O_3$ changes (Fig. 19) shows a nearly linear correlation for altitudes from the surface to 4 hPa with a tendency to saturation for larger $NO_x$ perturbations in the lower altitude emission scenarios. For altitudes from 4-0.01 hPa the correlation is curvilinear and the $NO_x$ emission scenarios show a larger range of values, compared to the lower altitude range. Here, the sensitivity is very large to the altitude and latitude of emission.

Since the correlation of $O_3$ and $NO_x$ perturbation is well known, we include the $O_3$ perturbation as a 'direct' effect. When comparing Fig. 20 and Fig. 6 the similarity is clearly visible. Areas of $NO_x$ increase overlap with areas of $O_3$ decrease. Additionally, we see areas of $O_3$ increase below areas of $O_3$ decrease at southern mid-latitude and northern tropic emission scenarios. Note that the magnitude is much smaller and results are clearly significant. For northern mid-latitude scenarios, these are barely visible in the tropics and for north polar emission scenarios they are not visible.




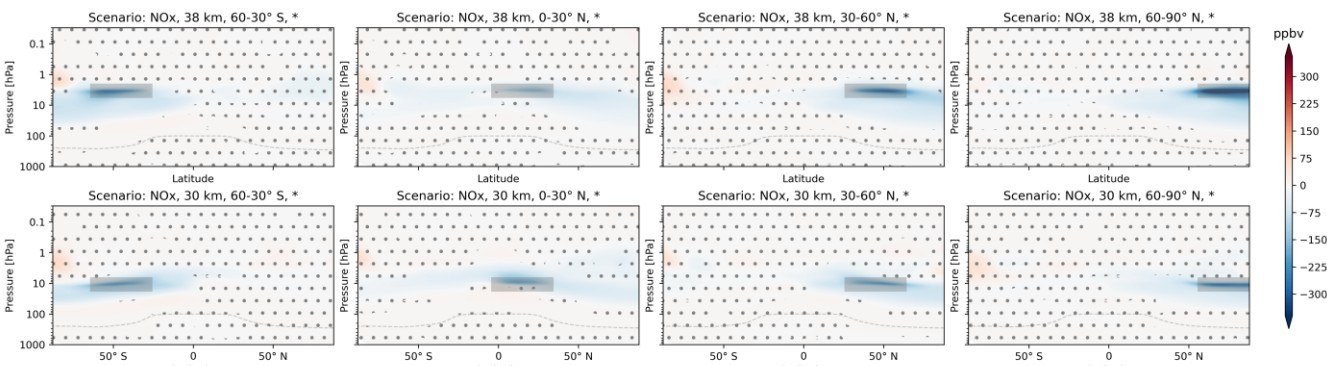

**Figure 6: Zonal mean of O₃ perturbation (volume mixing ratio, ppbv) for scenarios where NOₓ is emitted. The first and second row refer to 38 km and 30 km altitude of emission, respectively. Columns represent the latitude region of emission. The emission regions are shaded in grey. Dots indicate statistical insignificant results and the probability level is written in the title of each subplot.**

### 4.2.2. Indirect effects

Overall, indirect effects of NOₓ emission on $H_2O$, $H_2$ and $CH_4$ are basically insignificant for most areas. For some scenarios, significant areas appear for tropospheric and lower stratospheric altitudes, largely depending on which hemisphere NOₓ was emitted (Fig. 21, Fig. 22, Fig. 23).

### 4.3. Hydrogen emission

### 4.3.1. Direct effects

The increase in atmospheric concentrations of $H_2$by $H_2$ emission peaks at the emission location (Fig. 7). The perturbation pattern looks very similar to $H_2O$ perturbation from $H_2O$ emission and is therefore most probably dominated by transport, i.e. the Brewer-Dobson circulation, instead of photochemistry.

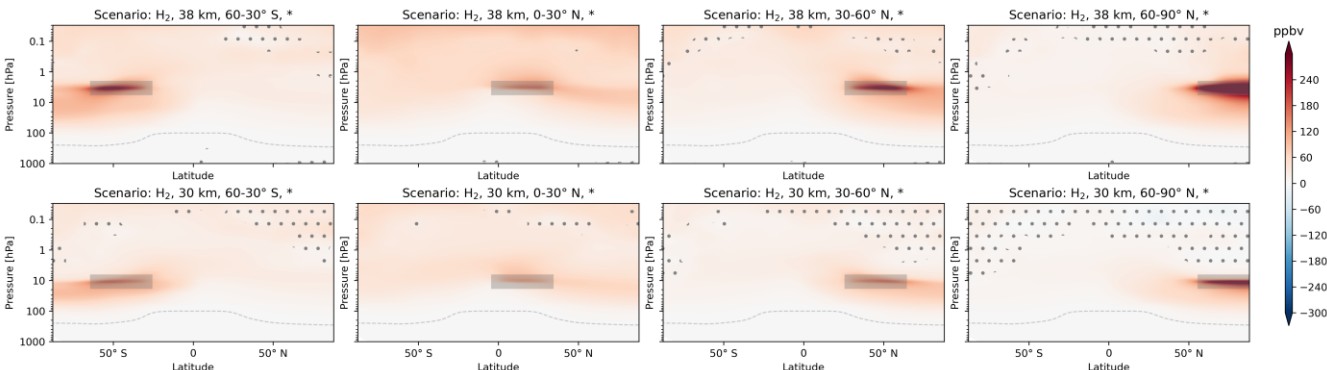

**Figure 7: Zonal mean of H₂ perturbation (volume mixing ratio, ppbv) for scenarios where H₂ is emitted. The first and second row refer to 38 km and 30 km altitude of emission, respectively. Columns represent the latitude region**



**of emission. The emission regions are shaded in grey. Dots indicate statistical insignificant results and the probability level is written in the title of each subplot.**

### 4.3.2. Indirect effects

The EMAC results show that $H_2$ emissions at 38 km generally and for the tropics at 30 km statistically significant reduce the
$O_3$ abundance in the upper stratosphere (0.1-1 hPa). It is known that above 40 km the HOx cycle starts to dominate the $O_3$ depletion instead of the $NO_x$ cycle (Zhang et al., 2021b; Matthes et al., 2022).

Maximum values of $H_2O$ perturbation are approximately 5 % of the $H_2O$ perturbation of the direct $H_2O$ emission scenarios. Apart from the magnitude, the features are very similar, which again suggests that transport dominates these perturbations (Fig. 25). Largest perturbations appear for the higher altitude scenarios and the low latitude scenarios. A correlation plot
shows that statistically significant $H_2O$ changes due to $H_2$ emission show different orders of magnitude depending on Hemisphere with larger gradients appearing rather in the Southern Hemisphere. Clearly, high latitude and low altitude emission scenarios show smaller $H_2O$ changes and again suggests a dominant role of transport for the perturbation lifetime.

$CH_4$ perturbation patterns are mostly not statistically significant (Fig. 26).

### 4.4. Methane, water vapour perturbation lifetime and overview of relative changes

### 4.4.1. Methane and methane lifetime

Tropospheric and whole model domain $CH_4$ lifetime of all emission scenarios is 8.39 and 9.54 years on average, respectively. We used the following equation to calculate both, the tropospheric and whole domain $CH_4$ lifetime $\tau_{CH4}$.

$$\tau_{CH_4}(t) = \frac{\sum_{b\epsilon B} M_{CH_4}^b(t)}{\sum_{b\epsilon B} k_{CH_4}^b(t) \cdot [OH]^b(t) \cdot M_{CH_4}^b(t)} \qquad (7)$$

$M_{CH_4}^b$ is the $CH_4$ mass, $[OH]^b$ the concentration and $k_{CH_4}^b$ the reaction rate of $CH_4$ + OH in grid box $b$. $B$ is the set of all grid
boxes.

Eq. (7) was applied to calculate the tropospheric $CH_4$ lifetime for different EMAC model setups (Fig. 16, Jöckel et al., 2016a). The results show an average value over all setups of 8.0±0.6 for the years 2000-2004, which, according to the authors, is on the lower end of a set of values from other publications.

Fig. 15 shows the $CH_4$ lifetime (a) and $H_2O$ mass perturbation (b) in relation to the relative hydroxyl radical mass change.
$CH_4$ lifetime changes of $H_2$ emission scenarios are quite close to the reference scenario and a clear trend is not visible. $H_2O$ emission scenarios show a clear correlation between OH increase and $CH_4$ lifetime decrease, with an increase or only small change at higher northern latitude and lower altitude scenarios and a decrease of $CH_4$ lifetime for southern and tropical




latitude scenarios (Table 3). An average over all emission scenarios per emission type shows an increase of relative hydroxyl and hydroperoxyl mass mostly in the middle atmosphere (Table 6). This explains the decreasing global $CH_4$ lifetime for $H_2O$

with a more efficient $CH_4$ oxidation due to an increase in OH (Fig. 15 (b)).

$NO_x$ emission scenarios, which show small OH perturbations close to zero, also show the largest reduction in $\tau_{CH4}$ with two altitude clusters (green shaded ellipses). As an addition, Fig. 16 in the appendix shows the tropospheric $CH_4$ lifetime, which shows very similar trends for $NO_x$ emission scenarios compared to the whole model domain $CH_4$ lifetime. According to literature two pathways of tropospheric chemistry connect $NO_x$ and OH concentrations. On one hand, $HO_2$ to OH recycling

is speed-up by NO and eventually increases OH. On the other hand, $NO_2$ oxidation reduces OH concentrations and increases nitric acid ($HNO_3$) concentrations (Ehhalt et al., 2015, p.p. 245). Note that tropospheric OH concentrations are only slightly increased for $NO_x$ emission scenarios (Table 6). The above-mentioned processes might not allow large perturbations of OH to build up, even though the effect on $CH_4$ lifetime are the largest in all scenarios.

In the troposphere, $H_2O$ emission and $NO_x$ emission scenarios show an inverse trend with a tropospheric $CH_4$ lifetime

increase (Fig. 16,) combined with tropospheric hydroxyl decrease (Table 6) for the former and tropospheric $CH_4$ lifetime decrease combined with tropospheric hydroxyl increase for the latter. In summary, the important processes for global $CH_4$ lifetime take place in the troposphere for $NO_x$ scenarios and in the middle atmosphere for $H_2O$ emission scenarios.

### 4.4.2. Water vapour perturbation lifetime

Fig. 8 shows an increase in $H_2O$ perturbation lifetime with altitude. Values at tropospheric altitudes range from

approximately 1 hour to half a year. Generally, the $H_2O$ perturbation lifetime is longest at tropical regions and high altitudes and gets less at higher latitudes and lower altitudes. The lifetime range at stratospheric altitudes is large from 1 month to 5.5 years, which includes the extension that is based on this work. We used our data to extend the altitude dependency of $H_2O$ perturbation lifetime in existing literature to higher altitudes. Grewe and Stenke (2008) published $H_2O$ perturbation lifetime from the surface to approximately 20 km (50 hPa). We extended the range up to approximately 40 km (3 hPa). The transport

of low latitude high altitude $H_2O$ emissions to the high latitudes of the troposphere along the shallow and deep branches of the Brewer-Dobson circulation takes the longest time. Hence, transport of $H_2O$ perturbations dominates the perturbation lifetime for both, the previous and the extended altitudes. Pletzer et al. (2022) showed that photochemistry does not reduce total $H_2O$ perturbations at 26 and 35 km cruise altitude. We report that perturbation lifetimes continues to increase with altitude for all latitudes and that photochemistry does not deplete total $H_2O$ perturbations for emissions up to 38 km.



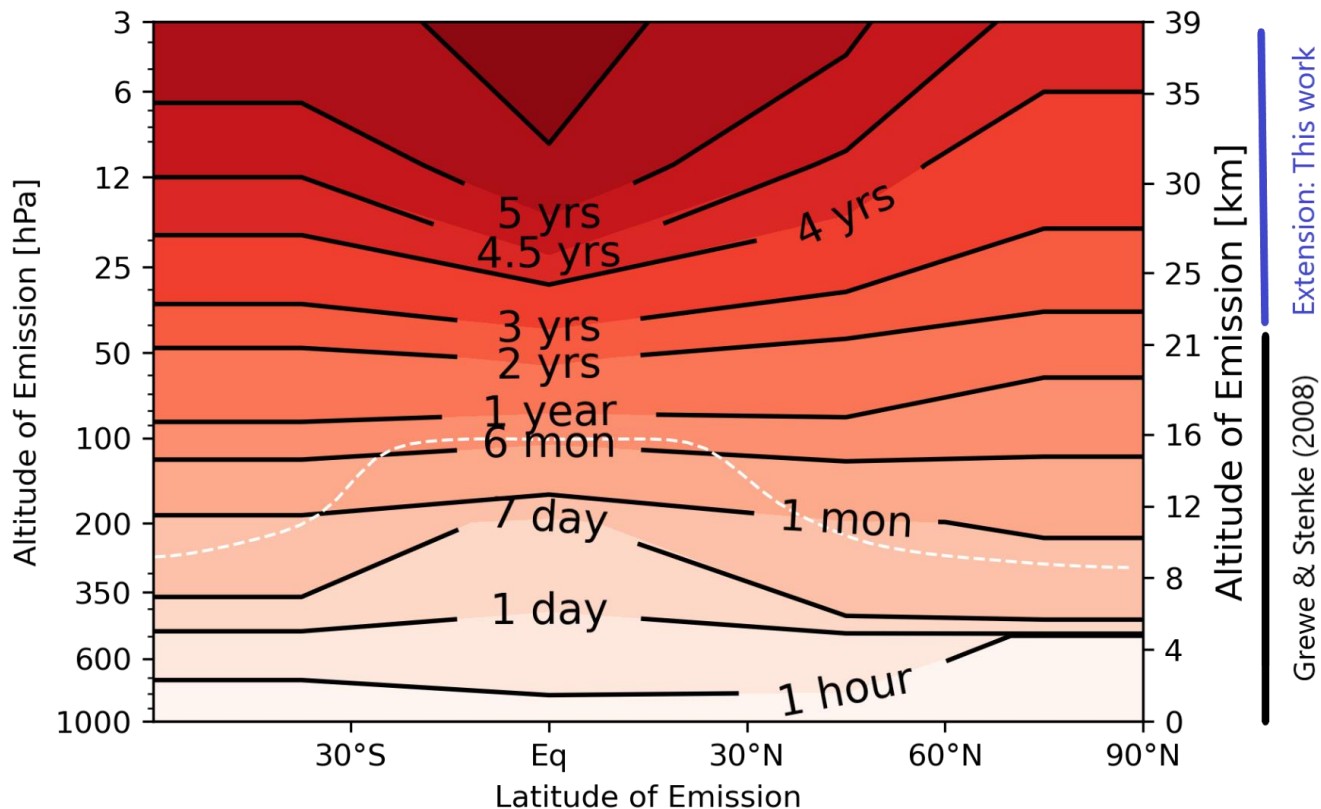


**Figure 8: Reproduced from Grewe and Stenke (2008), Fig. 6 (a), with permission from the authors, who provided the data. The figure shows the water vapour perturbation lifetime for the original pressure levels (1000-52 hPa) and the extended pressure levels (52-3 hPa) depending on latitude. The vertical regions are marked next to the right y-axis with black and blue lines, respectively.**

### 4.4.3. Overview on atmospheric composition

For a quick-look overview of sensitivities, we present Table 3, which shows the effect of one gigamole or teramole of $H_2O$, $NO_x$ and $H_2$ emission on $H_2O$ (above the tropopause), whole model domain $O_3$ and whole model domain $CH_4$ lifetime $\tau_{CH4}$. We added background colours blue, yellow and green, indicating that the values differ from zero with a probability of 95 %, 99.9 % and 99.99 %. The impact of $NO_x$, $H_2O$ and $H_2$ emission on $O_3$ is statistically significant with 99.99 % confidence for all 24 sensitivities. For sensitivities of $H_2O$ and $CH_4$ lifetime, 13 and 8 out of 24 means are different to the reference with at least 95 % confidence, respectively.



| Emission | | Sensitivity of Atmospheric Composition Changes to Emission of NO$_x$, H$_2$O and H$_2$ | | | | | | | | | | | | | | |
|---|---|---|---|---|---|---|---|---|---|---|---|---|---|---|---|---|
| | Altitude | ΔH$_2$O (above tropopause, WMO) | | | | Unit | ΔO$_3$ (Whole Model Domain) | | | | Unit | Δτ$_{CH4}$ (Whole Model Domain) | | | | Unit |
| 1 Gmol NOx | 30 km | -0,09 | 0,14 | -0,03 | -0,13 | Tmol | -0,17 | -0,14 | -0,20 | -0,18 | Tmol | -0,14 | -0,15 | -0,15 | -0,16 | % |
| | 38 km | 0,07 | 0,06 | 0,04 | 0,05 | Tmol | -0,12 | -0,12 | -0,16 | -0,17 | Tmol | -0,06 | -0,06 | -0,12 | -0,11 | % |
| 1 Tmol H$_2$O | 30 km | 4,58 | 5,44 | 4,39 | 3,42 | Tmol | -6,43 | -0,42 | -0,64 | -1,23 | x 10$^{-2}$ Tmol | -22,2 | -19,2 | 14,3 | 21,0 | x 10-3 % |
| | 38 km | 5,31 | 5,70 | 5,10 | 4,31 | Tmol | -6,36 | -3,21 | -0,69 | -1,22 | x 10$^{-2}$ Tmol | -29,6 | -41,9 | 1,2 | -2,1 | x 10-3 % |
| 1 Tmol H$_2$ | 30 km | 1,56 | 2,35 | 0,81 | 0,44 | Tmol | -7,4 | -15,1 | -12,2 | -8,7 | x 10$^{-2}$ Tmol | 10,9 | 0,9 | -12,6 | -0,8 | x 10-2 % |
| | 38 km | 2,86 | 3,69 | 2,45 | 1,62 | Tmol | -9,5 | -17,9 | -16,6 | -17,1 | x 10$^{-2}$ Tmol | 1,5 | -12,0 | -10,7 | 4,1 | x 10-2 % |
| | Latitude | 60-30° S | 0-30° N | 30-60° N | 60-90° N | | 60-30° S | 0-30° N | 30-60° N | 60-90° N | | 60-30° S | 0-30° N | 30-60° N | 60-90° N | |

| *** | ** | * |
|---|---|---|

**Table 3: Sensitivities of atmospheric composition changes, water vapour in teramole (ΔH₂O), relative ozone change in teramole (ΔO₃) and relative change of whole model domain methane lifetime (Δτ$_{CH4}$), to emission of H₂O, NO$_x$ and H₂ depending on altitude and latitude of emission.**

*4.4.3.1. Water vapour perturbation in the middle atmosphere*

The term stratospheric H$_2$O is quite common. Since our model includes parts of the mesosphere, we prefer to use the term mid-atmospheric H$_2$O. For H$_2$O emission, the sensitivity of mid-atmospheric H$_2$O increases clearly with altitude and is higher for lower latitude emission scenarios, which is also shown in Fig. 8. The impact of H$_2$ emission on mid-atmospheric H$_2$O is very similar and increases with altitude and is larger for lower latitudes. The order of magnitude of changes per molecule of emitted species shows that a molecule of H$_2$ is roughly 50 % as effective in enhancing the mid-atmospheric H$_2$O concentration as a molecule of emitted H$_2$O (Tbl. 3).

*4.4.3.2. Whole model domain ozone perturbation*

Generally, all three types of emissions cause O$_3$ depletion. However, the effect of NO$_x$ emission on whole model domain O$_3$ is two and three orders of magnitude larger compared to H$_2$ and H$_2$O emission, respectively. Hence, a NO$_x$ molecule is roughly 3 to 4 orders of magnitude more efficient in reducing the stratospheric O$_3$ burden than H$_2$ or H$_2$O. Interestingly, while in absolute values the H$_2$ emissions are of minor importance to the O$_3$ depletion, the average effectiveness in destroying O$_3$ is roughly 5-6 times larger for H$_2$ than for H$_2$O (Tbl. 3).

*4.4.3.3. Whole model domain methane lifetime change*

The number of significant results of CH$_4$ lifetime changes is low compared to atmospheric sensitivities of H$_2$O or O$_3$. CH$_4$ lifetime changes of NO$_x$ emission scenarios are lower at the higher altitude and show ranges between -0.06 to -0.16 % per Gmol of NO$_x$ emission. Per molecule CH$_4$ lifetime changes of H$_2$O emission scenarios are four to five orders of magnitude smaller than compared to NO$_x$ emission scenarios. Note that CH$_4$ lifetime in H$_2$O and NO$_x$ emission scenarios clearly shows a linear trend or clusters depending on altitude respectively, while H$_2$ emission scenarios do not show a clear trend (Fig. 15).





## 5. Radiative forcing

We used the atmospheric changes of the radiatively active gases, $H_2O$, $O_3$ and $CH_4$, to calculate the stratospheric-adjusted radiative forcing (total net RF) at the tropopause (about 180 hPa).

In Fig. 9, we present the total net RF, which shows the total shortwave (SW), total longwave (LW) and total net RF grouped by emission and aligned by latitude and marked by altitude of emission. Here, 'total' refers to the combined effect of RF due to $H_2O$, $O_3$ and $CH_4$ changes. In the following subsections, we address the individual, $H_2O$, $O_3$, $CH_4$ altitude and latitude dependencies of RF and the relation of RF to atmospheric composition changes.

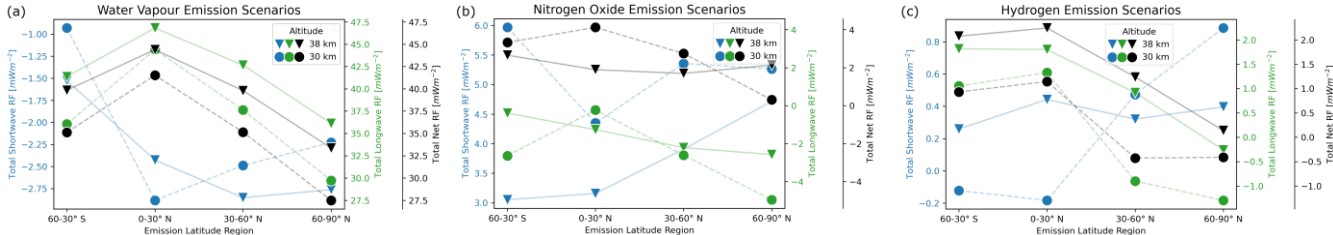

**Figure 9: The total shortwave, longwave and net radiative forcing in blue, green and black, respectively, for water vapour (a), nitrogen oxide (b) and hydrogen (c) emission scenarios due to atmospheric composition changes of water vapour, ozone and methane. For different altitude emission scenarios refer to markers. Latitude regions of emission are aligned on the x-axis.**

Comparing the magnitude, the $H_2O$ emission scenarios show the largest total net RF, followed by $NO_x$ and $H_2$ emission scenarios.

$H_2O$ emission scenarios all have a negative total SW RF (blue), which is smallest at southern mid-latitudes. The total LW RF (green) is largest for tropical, and smallest for the north polar emission scenarios. The high altitude emission scenarios show a larger total LW RF for all latitudes.

$NO_x$ emission scenarios show a positive total SW RF, which is larger for the lower altitude scenarios. In comparison, the total LW RF is negative and smaller, with no distinctive altitude dependency.

$H_2$ emission scenarios have a positive total SW RF, apart from the lower altitude emission scenarios at southern mit-latitudes and the tropics. The total LW RF clearly depends on altitude with larger values for the high altitude emission scenarios, the Southern Hemisphere and the northern tropics.

### 5.1. Water vapour radiative forcing

$H_2O$ emission scenarios have the largest radiative forcing of the emission scenarios, followed by $H_2$ and $NO_x$ emission scenarios. Fig. 28 shows the $H_2O$ SW RF, $H_2O$ LW RF and $H_2O$ net RF due to $H_2O$ changes.



For $H_2O$ emission scenarios the $H_2O$ SW RF has the largest negative values for emission in the tropics and the lowest negative values for emission at north polar latitudes. The $H_2O$ LW RF shows the inverse trend with largest positive values for emission in the tropics and lowest positive values for emission at north polar latitudes. Overall, $H_2O$ LW RF dominates the $H_2O$ net RF.

For $NO_x$ emission scenarios the $H_2O$ SW RF is mostly positive, whereas the $H_2O$ LW RF is negative. The $H_2O$ net RF is negative apart from the emission at 30 km in the tropics.

For $H_2$ emission scenarios the $H_2O$ SW RF is negative for emission at southern mid-latitudes and the tropics and positive for emission at most northern mid-latitudes and at the northernmost latitudes. The $H_2O$ LW RF shows the inverse trend with warming at southern mid-latitude and tropic emission and cooling only for emission at the lower altitudes for northern mid- and high-latitudes. The LW RF dominates the net RF by an order of magnitude.

**5.2. Ozone radiative forcing**

Fig. 29 is based on $O_3$ changes, excluding $H_2O$ and $CH_4$ changes. For $H_2O$ emission scenarios the $O_3$ SW RF is positive with maximum values of 1.75 mWm$^{-2}$for southern latitude emission and continuously becomes close to zero the further north $H_2O$ is emitted. The $O_3$ LW RF shows an inverted trend with large negative values for emission at southern latitudes and values between 0 and -1 mWm$^{-2}$ for other emission locations. The altitude dependency of $O_3$ SW RF is largest for emission at tropical regions and, in contrast, $O_3$ LW RF shows larger altitude differences only at higher latitude emission scenarios. Overall, there is a clear altitude distinction of net $O_3$ RF due to $H_2O$ emission and values are negative and positive.

For $H_2$ emission scenarios the $O_3$ SW RF shows only a small latitude dependency around 0.4 mWm$^{-2}$for the higher altitude emission scenarios. In contrast, the lower altitude emission causes negative values up to -0.2 mWm$^{-2}$for emission at southern latitudes to the tropics and has positive values for north polar emission. The $O_3$ LW RF is negative and differences of high and low altitude emission are largest for emission at the tropics. Since $O_3$ LW and SW RF tend to cancel each other, the net $O_3$ RF has generally lower values than the individual contributions and we report a stronger cooling by $O_3$ perturbation due to $H_2$ emission for the lower altitude at all latitude emission scenarios except north mid-latitude and north polar emission.

**5.3. Methane radiative forcing**

$CH_4$ composition changes contribute significantly less to RF than atmospheric composition changes of $H_2O$ and $O_3$. The range of $CH_4$ net RF is -0.17 to 0.05 mWm$^{-2}$ and LW RF is larger than SW RF in many scenarios (Fig. 30).



For $H_2O$ emission scenarios the $CH_4$ net RF cools more for the higher altitude scenarios. For lower altitude scenarios radiation flux changes are smaller, apart for northern mid-latitude emission with an effect close to zero and north polar emission with a comparably small heating.

The $NO_x$ emission scenarios show a warming for the lower altitudes scenarios, apart for emission at southern latitudes, for which a comparably larger cooling effect occurs. For higher altitude emission scenarios the $CH_4$ net RF is close to zero for southern mid-latitude and tropic emission and cools for northern mid-latitude and north polar emission of $NO_x$.

H2 emission scenarios show both warming and cooling with LW and SW due to the $CH_4$ perturbation. A continuous cooling appears for the higher altitude emission scenarios. The lower altitude emission scenarios show alternating values of warming and cooling depending on latitude.

## 5.4. Individual radiative forcing and the related perturbations

### 5.4.1. $H_2O$ net radiative forcing

The average sensitivity of $H_2O$ net RF to one teragram of $H_2O$ increase is 0.37±0.01 mW m$^{-2}$ (TgH$_2$O)$^{-1}$ for $H_2O$ emission scenarios, 0.10±0.79 mW m$^{-2}$ (TgH$_2$O)$^{-1}$ for $NO_x$ emission scenarios and 0.14±0.28 mW m$^{-2}$ (TgH$_2$O)$^{-1}$ for $H_2$ emission scenarios. The standard deviation, i.e. latitude and altitude variation, is large for the latter two and only $H_2O$ emission scenarios always result in a radiative warming with a low standard deviation.

Fig. 31 shows the relation of $H_2O$ net RF and $H_2O$ perturbation above the tropopause for all emission scenarios. The relation increases linearly for low values of $H_2O$ perturbation, with slight underestimate of $NO_x$ and $H_2$ emission scenarios. The results confirm the linear relation between stratospheric $H_2O$ perturbations and RF from earlier findings and for lower emission altitudes (dashed gray line, (Fig. 8, Grewe et al., 2014a)). However, for higher perturbations deviations in RF appear and the relation $\frac{\Delta RF}{\Delta H_2O}$ continuous to decrease. A two dimensional polynomial fit captures the data points well with very small deviations.

### 5.4.2. $O_3$ net radiative forcing

The average sensitivity of $O_3$ net RF to one percent of $O_3$ decrease is -9.9±38.2 mW m$^{-2}$ (%O$_3$)$^{-1}$ for $H_2O$ emission scenarios, 20.2±6.1 mW m$^{-2}$ (%O$_3$)$^{-1}$ for $NO_x$ emission scenarios and -7.4±8.7 mW m$^{-2}$ (%O$_3$)$^{-1}$ for $H_2$ emission scenarios. The interannual variability is particularly large for $H_2O$ emission scenarios. For the $NO_x$ emission scenarios, $O_3$ decrease always causes warming.

Fig. 32 shows the relation of $O_3$ net RF to relative $O_3$ change for all emission scenarios. The $NO_x$ emission scenarios are clustered and the related $O_3$ depletion causes warming. For the lower altitude scenarios, different levels of $O_3$ decrease do not



cause a large difference in warming. In contrast, for higher altitude scenarios the latitude of emission has a larger impact on
both $O_3$ depletion and $O_3$ net RF variability. $H_2O$ and $H_2$ emission scenarios are clustered around close to zero RF, apart
from three single $H_2O$ emission scenario values, where the pair is emission at southern mid-latitudes and the single value at
northern tropics. The comparably large $O_3$ depletion for $H_2O$ emission at southern mid-latitudes originates from enhanced
denitrification by increased $H_2O$ concentrations within polar stratospheric clouds (not shown), which is known to be stronger
in southern polar regions compared to northern polar regions (Tabazadeh et al., 2000).

### 5.4.3. CH₄ net radiative forcing

The average sensitivity of $CH_4$ net RF to one percent of global $CH_4$ decrease is -8.7±10.2 mW m$^{-2}$/% $CH_4$ for $H_2O$ emission
scenarios, -39.5±88.1 mW m$^{-2}$/% $CH_4$ for $NO_x$ emission scenarios and 2.1±35 mW m$^{-2}$/% $CH_4$ for $H_2$ emission scenarios.
Clearly, the spread is large.

All of $NO_x$ and $H_2$ emission scenarios are clustered around zero RF, with both radiative warming and cooling (Fig. 33). $H_2O$
emission scenarios show warming, no change and cooling for the lower altitude emission. For higher altitude emission the
larger $CH_4$ reduction comes with a cooling effect. Both altitudes combined scale approximately linear, however the line
crosses from cooling to warming slightly below the inversion of $CH_4$ decrease to $CH_4$ increase.

### 5.4.4. Summary on radiation

| Emission | | Sensitivity of Radiative Forcing caused by Atmospheric Composititon Changes to Emission of NOₓ, H₂O and H₂ | | | | | | | | | | | | | | | |
|---|---|---|---|---|---|---|---|---|---|---|---|---|---|---|---|---|---|
| | Altitude | ΔH₂O net RF | | | | Unit | ΔO₃ net RF | | | | Unit | ΔCH₄ net RF | | | | Unit |
| 1 Gmol NOx | 30 km | -0,54 | 1,20 | 0,05 | -1,02 | mWm⁻² | 5,84 | 5,98 | 4,21 | 3,22 | mWm⁻² | 0,009 | 0,028 | 0,023 | 0,021 | mWm⁻² |
| | 38 km | 0,84 | 0,45 | 0,57 | 0,43 | mWm⁻² | 4,73 | 2,22 | 3,08 | 3,42 | mWm⁻² | 0,018 | 0,028 | -0,020 | -0,022 | mWm⁻² |
| 1 Tmol H₂O | 30 km | 30,66 | 34,19 | 29,74 | 24,07 | mWm⁻² | -3,60 | -1,49 | -2,14 | -2,15 | mWm⁻² | 0,016 | 0,005 | 0,028 | 0,010 | mWm⁻² |
| | 38 km | 34,09 | 35,59 | 32,78 | 28,67 | mWm⁻² | -3,38 | -1,28 | -1,66 | -2,30 | mWm⁻² | 0,012 | -0,005 | -0,003 | -0,018 | mWm⁻² |
| 1 Tmol H₂ | 30 km | 10,45 | 15,67 | 6,01 | 3,27 | mWm⁻² | -6,14 | -4,96 | -4,31 | -3,53 | mWm⁻² | 0,324 | 0,230 | -0,146 | 0,300 | mWm⁻² |
| | 38 km | 18,42 | 24,16 | 16,85 | 11,10 | mWm⁻² | -7,71 | -5,22 | -5,49 | -4,96 | mWm⁻² | -0,005 | -0,064 | 0,003 | 0,229 | mWm⁻² |
| | Latitude | 60-30° S | 0-30° N | 30-60° N | 60-90° N | | 60-30° S | 0-30° N | 30-60° N | 60-90° N | | 60-30° S | 0-30° N | 30-60° N | 60-90° N | |

*** ** *

**Table 4: Sensitivities of three net radiative forcings (H₂O, O₃, CH₄) to emission of H₂O, NOₓ and H₂ depending on altitude and latitude of emission.**

Table 4 shows an overview of radiative sensitivities normalized to perturbation per emitted mole. The error potential is
labeled according to the t-test, that was calculated for atmospheric composition changes and the related probabilities are
listed in Tbl. 2. The $H_2O$ net RF of $NO_x$ emission scenarios is statistically not significant according to the t-test. Note that the
normalization to perturbation per emitted mole has a different magnitude for $NO_x$ emission scenarios (gigamole). In contrast
to $NO_x$ emission scenarios, the $H_2O$ net RF for $H_2O$ emission scenarios are all statistically significant. Note that $H_2O$
emission scenarios would contribute by far the most to net radiative forcing without the normalization, followed by $NO_x$ and





$H_2$ emission scenarios, which is mainly due to the large ratio of $H_2O$ to $NO_x$ and $H_2$ in the exhaust. The normalized $O_3$ net RF is significant for all three types of emission. $NO_x$ emission scenarios show by far the largest $O_3$ net RF values, followed

by $H_2$ and $H_2O$ emission scenarios (if the order of magnitude is kept in mind). Normalized $CH_4$ net RF is significant for most $NO_x$ emission scenarios, however the magnitude is small compared to the normalized $O_3$ net RF for $NO_x$ emission.

## 6. Discussion of atmospheric composition changes and radiative forcing

This section addresses the relation of emission, atmospheric composition changes and radiative forcing. A variety of publications exist where idealized atmospheric composition changes are used for sensitivity studies of the radiative effect

Lacis et al. (1990a);Hansen et al. (1997a);de F. Forster and Shine (1997), where Riese et al. (2012) is one of the most recent ones. However, our approach adds a level of complexity, since we did not calculate the RF of idealized atmospheric composition changes, but of modelled atmospheric composition changes due to idealized emission scenarios. Hence, it is important to discuss the process of emission, followed by atmospheric composition changes and the radiative forcing.

### 6.1. Water vapour atmospheric and radiative sensitivities

In the discussion of $H_2O$ changes and their radiative forcing, we exclude $NO_x$ emission scenarios, since they show no statistically significant results (see Table 3 and Table 4). Middle atmospheric $H_2O$ perturbation for $H_2O$ and $H_2$ emission scenarios depends very much on altitude of emission and increases with altitude of emission. The $H_2O$ perturbation lifetime follows more or less the Brewer-Dobson circulation, where $H_2O$ emitted into the uprising tropical air has a larger lifetime than $H_2O$ into the sinking polar air. In contrast to the increase with altitude, $H_2O$ net RF develops a curvilinear trend and

begins to decrease with mass perturbation (Fig. 31). The effect is small compared to the total perturbation. Two possible explanations are, first, a saturation of reflected $H_2O$ longwave radiation, and, second, altitude differences in peak $H_2O$ mass accumulation with smaller radiative sensitivities for higher altitudes (Riese et al., 2012). In a prior publication (Pletzer et al., 2022), we did test $H_2O$ net RF according to Myhre et al. (2009) and we deem a dependency of $H_2O$ net RF to stratospheric $H_2O$ background less likely. Fig. 34 shows vertical profiles of globally integrated $H_2O$ concentration changes, which

supports the second explanation in our opinion. To explain, excluding Southern Hemisphere scenarios, the peak accumulation is at higher altitudes for scenarios where the trend of $H_2O$ net RF deviates more from the linear trend in Fig. 31. Hence, an altitude and or latitude shift of the main mass accumulation to higher values should cause the lower $H_2O$ net RF. Grewe et al. (2014a);Wilcox et al. (2012) reported a nearly linear dependency of stratospheric adjusted RF to emission magnitude for a range of 9.5-11.5 km. Apparently, there the radiative sensitivity to different magnitudes does not show

curvilinear tendencies. In contrast, the radiative sensitivity shows a curvilinear trend around the tropopause (Fig. 2b, van Manen and Grewe, 2019). To summarize briefly, two trends of atmospheric composition changes and radiative forcing oppose each other. First, the increase in $H_2O$ perturbation with altitude and, second, the decreasing radiative sensitivity



depending on altitude and latitude. The first dominates the $H_2O$ net RF and the second is a second order variation of the $H_2O$ net RF.

**6.2. Ozone atmospheric and radiative sensitivities**

All scenarios show a total $O_3$ depletion, however, also increases in $O_3$ at various altitudes (Fig. 6, Fig. 4, Fig. 24). Hence, regions of $O_3$ depletion are partly equilibrated by regions of $O_3$ increase in terms of total depletion. Clearly, the perturbation sensitivity to emission is complex. The radiative sensitivity further increases the complexity. Close to the tropopause and the tropics the radiative effect per unit mass change is large (Fig. 1, Riese et al., 2012). In addition, several authors reported $O_3$ climate sensitivities, where an increase in $O_3$ either cools or warms near-surface air depending on domain (Fig 1, Lacis et al., 1990b; Fig. 7, Hansen et al., 1997b). This inversion point is slightly below 30 km, above which an increase in $O_3$ causes cooling of near-surface air. Even though the emission altitudes in this publication are 30 and 38 km and hence above the inversion point, the regions of $O_3$ increase or decrease are very much distributed at different altitude and latitude regions, where each region has its specific radiative sensitivity. Therefore, the regions with $O_3$ changes combined with the active radiative sensitivity there form a complex net total of warming and cooling. Since the $O_3$ net RF is positive for all $NO_x$ emission scenarios, we expect to see either $O_3$ depletion and hence warming at high altitudes and $O_3$ increase and hence warming at lower altitudes.

For presentation of stratospheric trace gas changes volume mixing rations are often preferred, e.g. because they are unaffected by transport processes. However, radiation impacts are mainly affected by density changes and this might change the point of view. For example, perturbations close to the tropopause might appear low as mixing ratios, but are larger in numbers due to higher air densities compared to higher altitude mixing ratio changes. Fig. 35 gives a direct comparison of profiles of density and volume mixing ratio changes. Many $NO_x$ emission scenarios (Fig. 35, mid) show an increase in $O_3$ density from 9 km (300 hPa) upwards, which switches to a decrease between around 16 and 25 km (100 and 25 hPa) depending on the specific emission scenario. To summarize, while the total density increase is smaller than the total density decrease, the region of increase comes with a significantly larger radiative sensitivity, which explains the radiative warming associated with reported total $O_3$ depletion in the scenarios. For $H_2O$ and $H_2$ emission scenarios regions of $O_3$ depletion and $O_3$ increase in combination with radiative sensitivities cause cooling in contrast to $NO_x$ emission scenarios. Here, the distribution of perturbations in combination with radiative sensitivities results in the opposite result to $O_3$ changes by $NO_x$ emissions.

In summary, both the radiative sensitivity and the location of perturbation patterns along latitude and altitude are crucial for the interpretation of the results. The radiative sensitivity is generally largest at the tropics and close to the upper-troposphere lower-stratosphere, while the perturbation patterns are complex and differ depending on emitted trace gas.



## 7. General discussion

In the previous sections we concentrated on the main sensitivities of hypersonic emissions. Processes that are generally
important, that with respect to the effect of hypersonic emissions are only of secondary importance, are discussed here. This
comprises the discussion of polar stratospheric clouds and heterogeneous chemistry and the net production of $H_2O$ from
hypersonic aircraft emissions. We include a comparison to results from literature about atmospheric impacts of supersonic
aircraft and synergy effects of simultaneous emissions.

### 7.1. Polar stratospheric clouds

Throughout this publication we focused on homogeneous, i.e. gas phase, atmospheric composition, since most of the
emissions affect atmospheric regions outside (spatial and temporal) polar night and spring processes. Our model setup
includes heterogeneous chemistry, i.e. particle effects like nucleation or condensation, which play a major role in polar
stratospheric clouds, as well as chlorine and bromine activation of those particles. There are two reasons, why this is
important for hypersonic aircraft emission. First, sedimentation of nitric acid trihydrate (NAT) and ice particles transport
both, $HNO_3$ and $H_2O$, from high to lower altitudes (Iwasaka, Yasunobu and Hayashi, Masahiko, 1991; Crutzen and Arnold,
1986), which effectively increases denitrification and dehydration and in turn could reduce perturbation lifetimes. Clearly,
the effect only contributes to the vertical transport, which is dominated by the residual circulation in the middle atmosphere.
Second, the chemistry within polar stratospheric clouds, where unreactive chlorine becomes reactive, heavily depletes $O_3$
concentrations and is prolonged by denitrification (Pyle, 2015, p.p. 248). In our model results, $HNO_3$ mixing ratios are
increased between 100-10 hPa for $NO_x$ emission scenarios. For $H_2O$ and $H_2$ emission scenarios the mixing ratios are
increased at around 10 hPa, depleted between 100-10 hPa and increased at and below 100 hPa (not shown). The effect is
significant in most regions for the former and to a lesser extent for the latter. Sedimentation change of $HNO_3$ (excluding $H_2$
emission scenarios) and ice appears at 10-100 hPa, with a peak between 10-20 hPa and is increased particularly in the lower
polar stratosphere at approximately 200 hPa, but peaks not appear below the tropopause. According to Iwasaka, Yasunobu
and Hayashi, Masahiko (1991) only NAT particles grown in the upper polar stratospheric clouds can reach the troposphere
and ice particles are the ones that evaporate in the lower stratosphere. In our model the vertical falling distance is defined by
the sedimentation velocity, which depends on the mean radius and a sedimentation factor (Kirner et al., 2011b). Hence, the
emitted trace gases, which become part of polar stratospheric cloud chemistry and sedimentation, should not become large
particles, since they do not reach the troposphere. In summary, polar stratospheric clouds affect atmospheric composition by
an enhanced vertical transport, which in turn affects nitrogen oxide concentrations to a larger extent compared to $H_2O$
concentrations. The effect becomes obviously more important for emission at high latitudes.



## 7.2. Net production of water vapour

Pletzer et al. (2022) showed that the emission of $H_2O$, $H_2$ and $NO_x$ result in a net-production of $H_2O$, which overcompensates the $H_2O$ depletion and increases the initial annual emission to some extent. In the study the origin of the effect, whether it comes from $H_2O$, $H_2$ or $NO_x$ emission, was not clear. In our simulation setup with independent emissions, we see that the overcompensation is largest for $H_2O$ and $H_2$ emission scenarios and to a small extent appears in $NO_x$ emission scenarios (not shown). The overcompensation is driven by oxidation in all scenarios and particularly an enhanced $HNO_3$ and $CH_4$ oxidation are very important. $H_2$ oxidation has a larger role in $H_2$ emission scenarios only.

## 7.3. Comparison to current literature on supersonic aircraft

The sensitivities to emissions of supersonic aircraft have recently been reviewed by Matthes et al. (2022). They describe the effect of $NO_x$ on $O_3$ and state that an inversion point of emission exists at 17 km below which emission of $NO_x$ increases total $O_3$ (Zhang et al., 2021c). Hypersonic aircraft cruising at 30 or 38 m show no clear picture and depletion of $O_3$ decreases or increases depending on latitude of emission with altitude. The importance of $H_2O$ emissions of supersonic aircraft and their effect on $O_3$ are highlighted, which we can further underline, since the total $O_3$ depletion due to $H_2O$ emission is sometimes only a factor different to $NO_x$ emission depending on scenario and the decreasing $O_3$ depletion with altitude for $NO_x$ emission scenarios is compensated if combined with $H_2O$ emission scenarios. Note that this depends very much on the ratio of emitted $NO_x$ and $H_2O$ and may easily change for a different ratio of exhaust. Matthes et al. (2022) further report that supersonic aircraft affect the $CH_4$ lifetime through $O_3$ changes and increased OH availability and an increased UV radiation in the troposphere. According to them the effect is comparably less important than the others. However, as shown in Pletzer et al. (2022), according to the trend calculated by both their models, $CH_4$ radiation changes might be more important for hypersonic aircraft compared to supersonic aircraft, where cooling by $CH_4$ changes compensates approximately a third of warming by $O_3$ changes (LMDZ-INCA model). In this study total $CH_4$ depletion for $H_2O$ emission scenarios clearly increases with altitude and hence might the cooling associated with it, but open questions on the order of magnitude of radiative forcing by $CH_4$ changes remain.

## 7.4. Synergy effects of simultaneous emissions

In our model and simulation setup we look at each emission independently. For comparison, Kinnison et al. (2020b) calculated single ($H_2O$, $NO_x$) and simultaneous emission, which we decided against for two reasons. First, due to their results - we have the impression that their single emission perturbations add up to the simultaneous emission perturbation (Fig. 6-8, Kinnison et al., 2020b) - and, second, to avoid a substantial increase in computation time. The results of this publication (e.g. Fig. 31) are very similar in magnitude compared to the combined emission calculated in Pletzer et al. (2022). Hence, effects due to simultaneous emission should at most be second order effects with small impact in comparison to the total effect.



### 7.5. Comparison to land hydrogen emissions

Ocko and Hamburg (2022) combined multiple studies and made a comparison of (indirect) radiative efficiencies or in our words radiative sensitivities to $H_2$ emission with other land emissions. Here, 'indirect' refers to tropospheric $CH_4$ and $O_3$
changes and stratospheric $H_2O$ changes due to $H_2$ emissions. Recalculated, their sensitivities to land $H_2$ emission amounts to 0.7 and 1.1 mW(m · TmolH$_2$)$^{-1}$, where the former excludes and the latter value includes the stratospheric effects. In this comparison, the radiative sensitivities to hypersonic aircraft $H_2$ emissions (Table 4), which are emitted directly in the middle atmosphere, are up to one order of magnitude larger than the radiative sensitivity to land $H_2$ emission.

### 8. Summary

In this study we analyzed sensitivities with respect to location and emission type of hypersonic emissions, and showed, first, how emissions of hypersonic aircraft ($H_2O$, $NO_x$, $H_2$) affect atmospheric composition and, second, how this change in atmospheric composition affects climate, i.e. stratospheric-adjusted radiative forcing. The novelty here is the systematic emission at two different altitude regions (30 km, 38 km) and four different latitude regions (60-30° S, 0-30° N, 30-60° N, 60-90° N) and the individual impact of $NO_x$, $H_2O$ and $H_2$ emissions.

Atmospheric perturbations were calculated with the full-scale atmospheric chemistry and general circulation model EMAC and the perturbations were fed to a radiation model to calculate the stratospheric-adjusted radiative forcing. Additionally, the study includes an evaluation of EMAC $H_2O$ and $O_3$ mixing ratios. Briefly summarized, the model excels in modelling $O_3$ and underestimates $H_2O$ mixing ratios in the middle atmosphere. The model setup was based on a novel approach to reduce the cost of computation and effectively reduced the simulated years by one-third.

The method to calculate changes in atmospheric composition and associated stratospheric-adjusted radiative forcings for different emission regions allows a comparison of their specific sensitivities. The main results are condensed in two quick-look tables of atmospheric and radiative sensitivities (Table 3, Table 4), visualized in Fig. 10, and other results are mostly for explanation and analysis. The main message is that sensitivities can differ manifold depending on latitude and altitude of emission. The most important large-scale process controlling the lifetime of perturbations certainly is the Brewer-Dobson
circulation. But also local processes like polar stratospheric cloud chemistry can contribute strongly depending on emission location. From the calculated sensitivities $H_2O$ emission and the related $H_2O$ perturbation in addition to $NO_x$ emission which causes $O_3$ perturbation have the largest warming potential. Both share the regional pattern (except $NO_x$ emission at 38 km), that emission at low latitudes is associated with the largest radiative sensitivities. However, their altitude sensitivities are (mostly) inverse, i.e. warming from $H_2O$ perturbation increases with altitude, while warming from $O_3$ changes decreases
with altitude at 30-38 km. It might appear in Fig. 10 that regions of largest water vapour changes in mid-latitudes and polar





regions do not overlap very much with the regions of largest radiative sensitivities in the tropics. However, the overlap clearly suffices to cause significant values of stratospheric-adjusted radiative forcing. The calculated sensitivities allow inexpensive and fast estimates of the stratospheric-adjusted radiative forcing of new hypersonic aircraft designs depending on latitude, altitude and ratio of emissions without the need to apply a complex atmospheric chemistry general circulation 735 model.

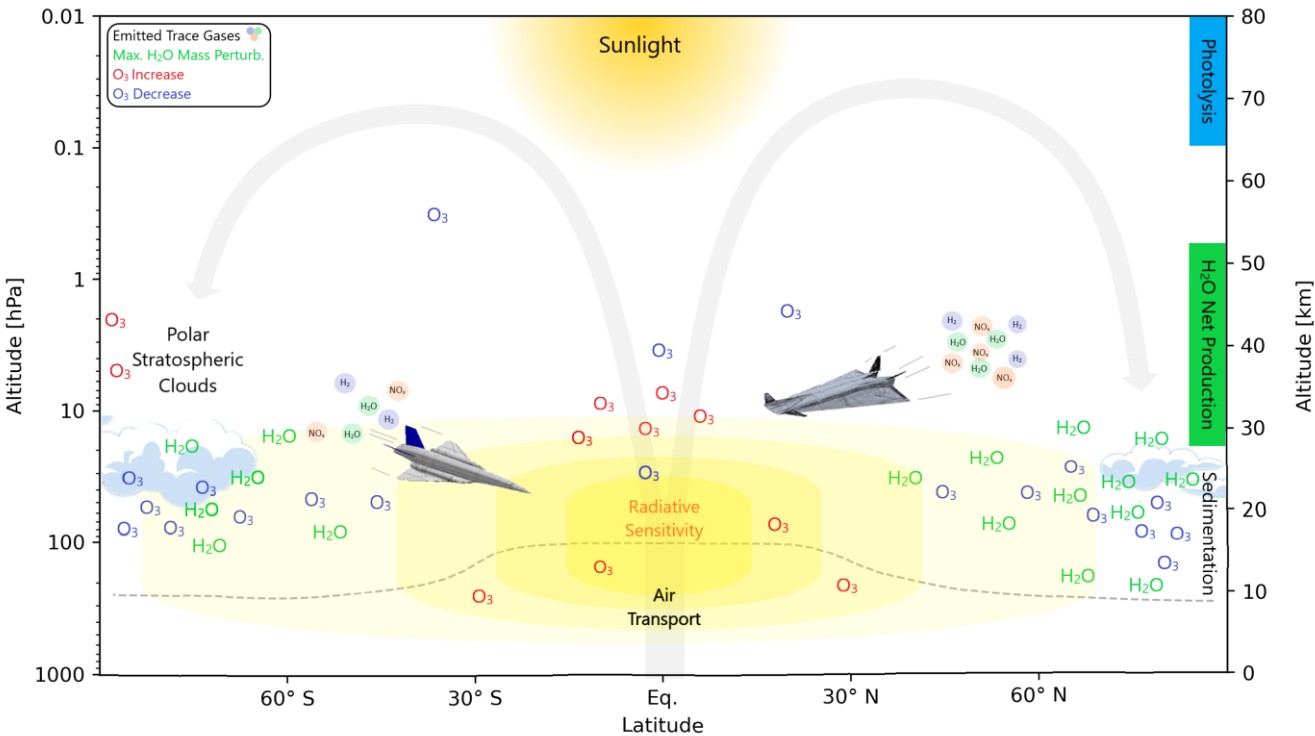

**Figure 10: Overview of the average atmospheric sensitivities of water vapour and ozone to hypersonic emissions of water vapour, nitrogen oxides and hydrogen and the most important atmospheric processes. Show in yellow is the approximate radiative sensitivity to trace gas changes.**

Generally, the results are in line with prior publications (Kinnison et al., 2020a; Pletzer et al., 2022) and the altitude optimization potential has already been highlighted by Kinnison et al. (2020a) for atmospheric composition changes and Pletzer et al. (2022) for atmospheric composition changes and radiation. An additional highlight is an extension of the $H_2O$ perturbation lifetime in literature to higher levels (Fig. 8). Further work may include the analysis of our simulation data with respect to the seasonal sensitivity that may enhance the mitigation potential when adapting aircraft trajectories to the 745 seasonal changes in circulation and chemistry.



## 9. Code availability

The Modular Earth Submodel System (MESSy) is continuously further developed and applied by a consortium of institutions. The usage of MESSy and access to the source code is licensed to all affiliates of institutions that are members of the MESSy Consortium. Institutions can become a member of the MESSy Consortium by signing the MESSy Memorandum of Understanding. More information can be found on the MESSy Consortium Website (http://www.messy-interface.org, last access: 14 July 2023). The submodel H2OEMIS used here has been implemented in MESSy version 2.54.0 and is available in the current devel branch

## 10. Data availability

For access to datasets of EMAC results please contact Johannes Pletzer.

## 11. Appendices

### 11.1. Methods and simulations

### 11.1.1. EMAC model setup

Additional diagnostics were activated via the following submodels: CONTRAIL, DRADON (Decay RADioactive ONline), O3ORIG (O3 ORIGin), ORBIT, PTRAC (Passive TRACer), SATSIMS (Satellites Simulator), SCALC (Simple CALCulations), TBUDGET (Tracer BUDGET), TENDENCY, S4D (Sampling in 4 Dimensions), SCOUT (Stationary Column OUTput), SORBIT (Satellite ORBIT), TROPOP (TROPOsPhere) and VISO (Vertically layered ISO-surfaces and maps). We mainly used PTRAC for verification of emitted trace gases, TENDENCY to account for and verify the specific humidity budget and TROPOP for the global WMO tropopause height during post-analysis.



## 11.1.2. Evaluation with satellite data

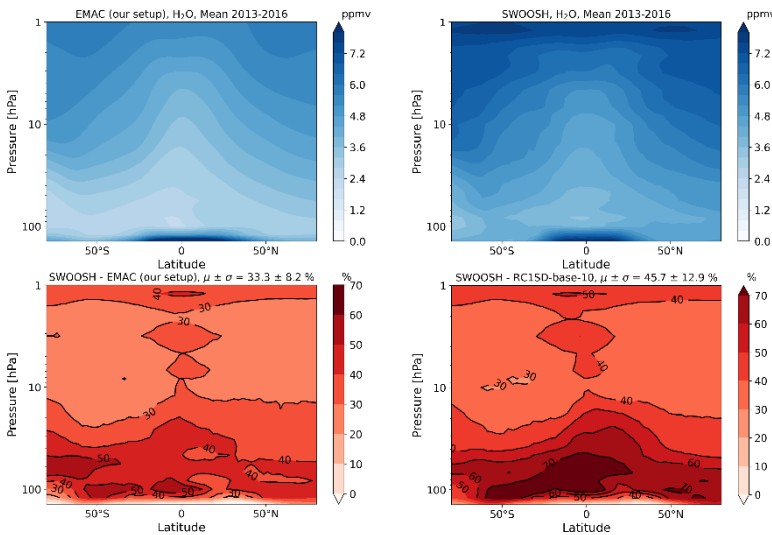

**Figure 11: Multi-annual mean (2013-2016) of water vapour volume mixing ratio for the EMAC setup, used in this work (upper left), SWOOSH multi-instrumental mean satellite data (upper right) and the difference (observation-model) in percent for our model setup (lower left) and RC1SD-base-10 (lower right).**

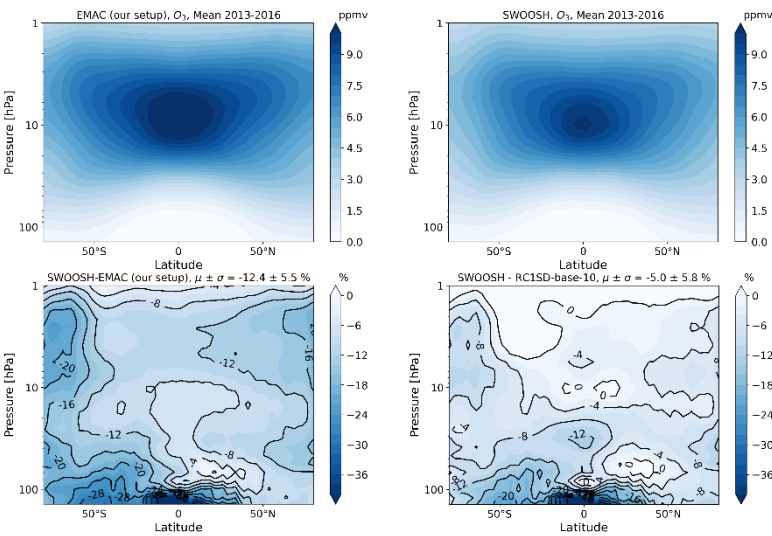

**Figure 12: Multi-annual mean (2013-2016) of ozone volume mixing ratio for the EMAC setup, used in this work (upper left), SWOOSH multi-instrumental mean satellite data (upper right) and the difference (observation-model) in percent for our model setup (lower left) and RC1SD-base-10 (lower right).**



### 11.1.3. Enhancing the efficient use of computing resources

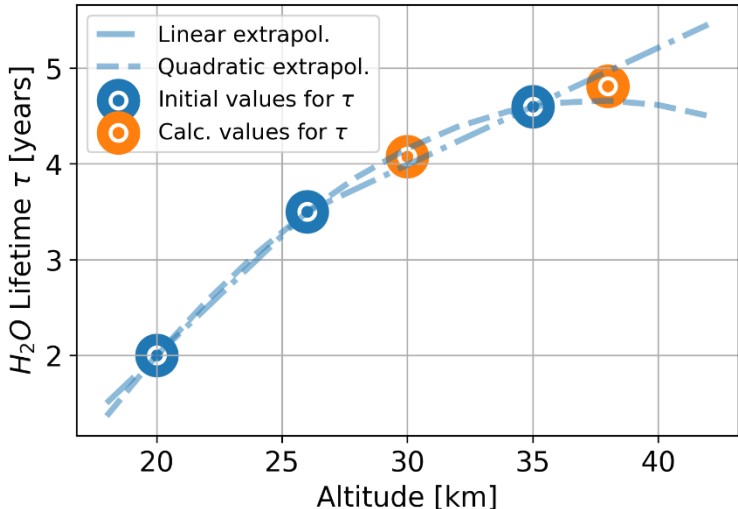

**Figure 13: Water vapour perturbation lifetime $\tau$ depending on altitude. Blue circles represent initial data, that were used for inter- and extrapolation of orange data circles. Dashed and dashed-dotted blue lines show the behavior of quadratic and linear inter- and extrapolation, respectively. Orange data circles are based on the average of linear and quadratic inter- and extrapolation. The values of the orange circles were used to calculate $s$ with Eq. (6).**



## 11.2. Emission scenarios

### 11.2.1. Timeline and validation

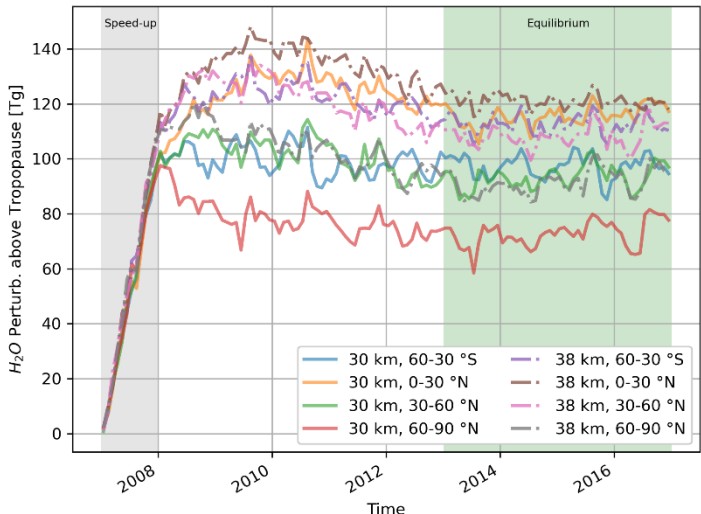

**Figure 14: Timeline of monthly mean water vapour perturbation in teragram for scenarios, where water vapour is emitted. The first year (gray shaded area) shows the enhanced emission by the factor _s_, i.e. the speed-up. 2013-2016 (green shaded area) shows the years in multi-annual mean equilibrium. The white and gray shaded area (2007-2012) marks the spin-up phase.**



## 11.3. Atmospheric composition changes

### 11.3.1. Methane lifetime

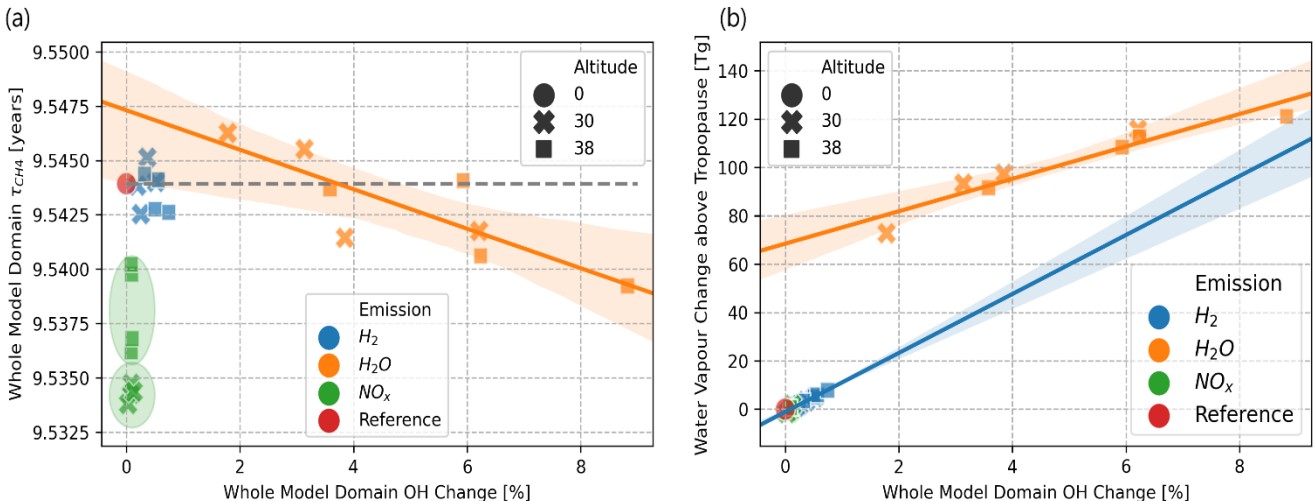

**Figure 15: Methane lifetime (whole model domain) is shown in (a) and the water vapour mass perturbation (b) in relation to the relative hydroxyl radical mass change (whole model domain) for all emission scenarios (legend). Whole model domain methane lifetime is reduced or extended below or above the dashed gray line, respectively. The regression lines including the shaded areas depict the mean and standard deviation of scenarios.**





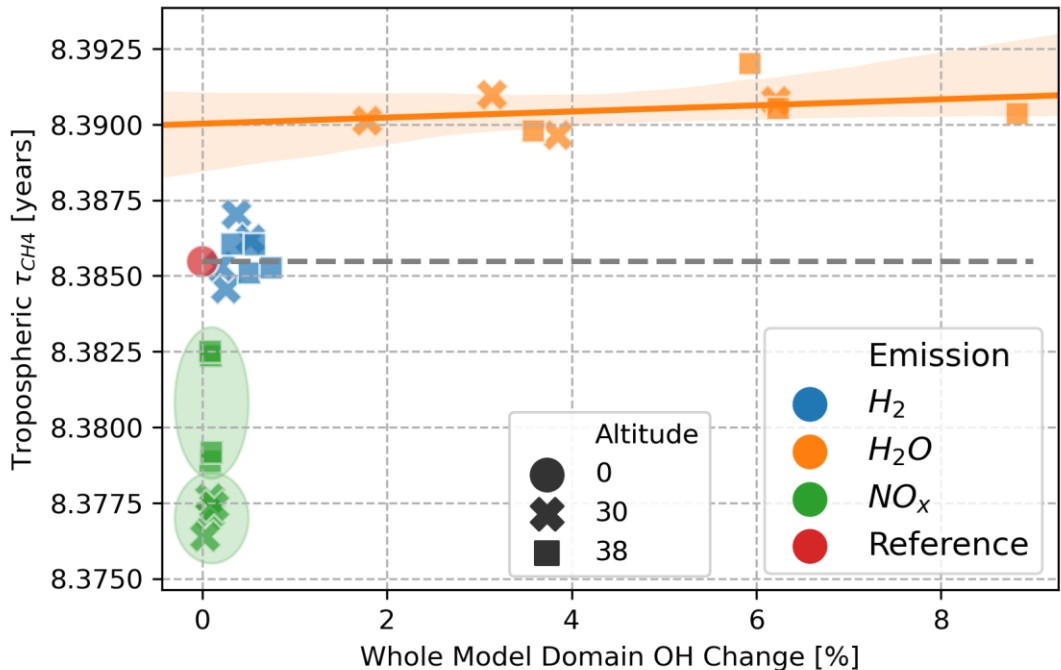

**Figure 16: Tropospheric methane lifetime in relation to hydroxyl change (whole model domain) for all emission scenarios (legend). Tropospheric methane lifetime is reduced or extended below or above the dashed gray line, respectively. The regression lines including the shaded areas depict the standard deviation of scenarios.**

### 11.3.2. Water vapour emission

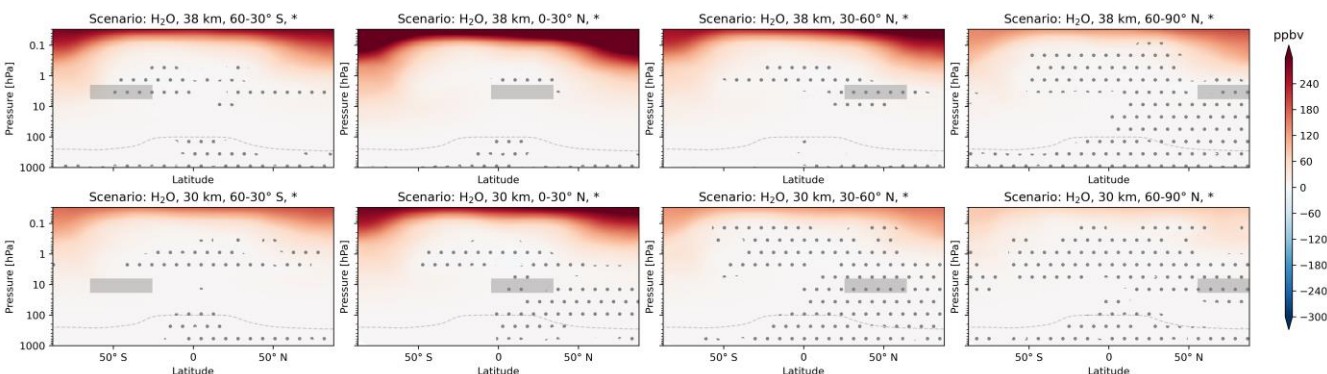

**Figure 17: Zonal mean of $H_2$ perturbation (volume mixing ratio, ppbv) for scenarios where $H_2O$ is emitted. The first and second row refer to 38 km and 30 km altitude of emission, respectively. Columns represent the latitude region of emission. The emission regions are shaded in gray. Dots indicate statistical insignificant results and the probability level is written in the title of each subplot.**



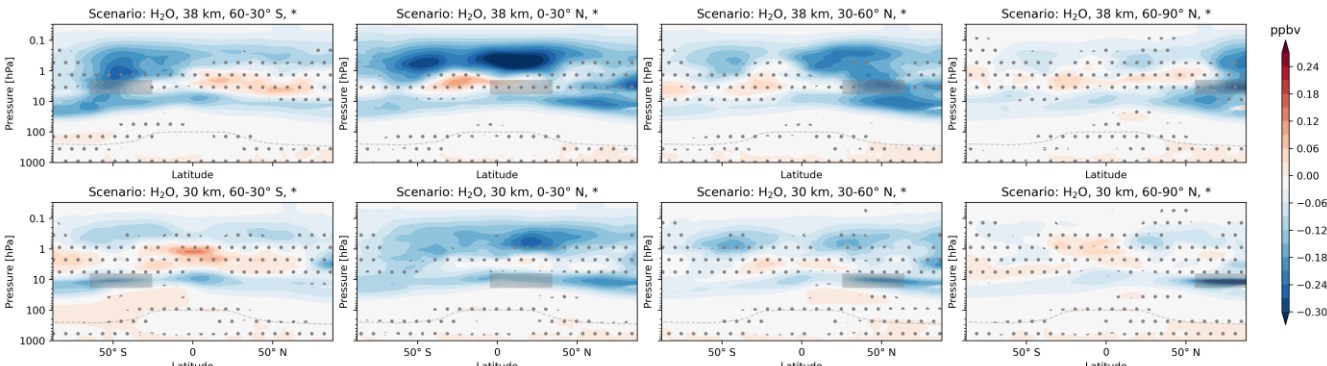

**Figure 18: Zonal mean of NOₓ perturbation (volume mixing ratio, ppbv) for scenarios where H₂O is emitted. The first and second row refer to 38 km and 30 km altitude of emission, respectively. Columns represent the latitude region of emission. The emission regions are shaded in gray. Dots indicate statistical insignificant results and the probability level is written in the title of each subplot.**

### 11.3.3. Nitrogen oxide emission and ozone perturbation

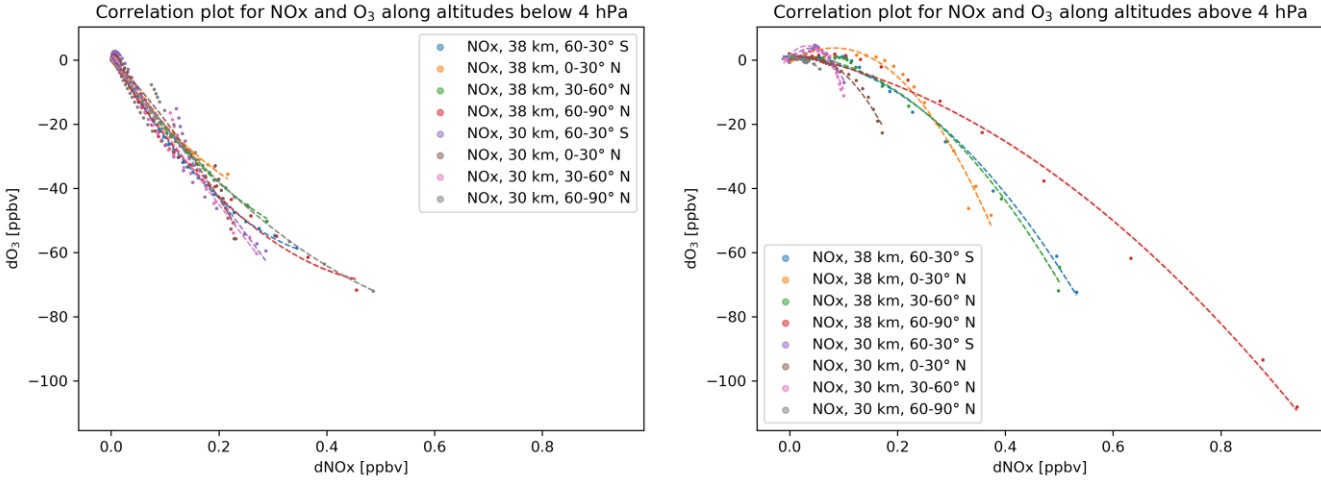

**Figure 19: Correlation of nitrogen oxide change and ozone change for two altitudes regions (surface-4 hPa and 4-0.01 hPa). The values are averaged over latitude and limited to statistically significant values with a 95 % confidence.**



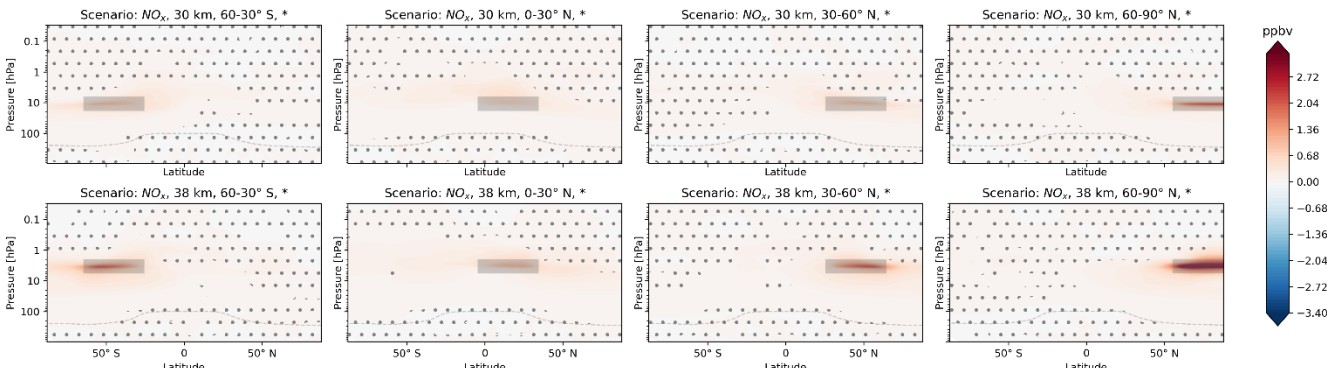

**Figure 20: Zonal mean of NO$_x$ perturbation (volume mixing ratio, ppbv) for scenarios where NO$_x$ is emitted. The first and second row refer to 38 km and 30 km altitude of emission, respectively. Columns represent the latitude region of emission. The emission regions are shaded in grey. Dots indicate statistical insignificant results and the probability level is written in the title of each subplot.**

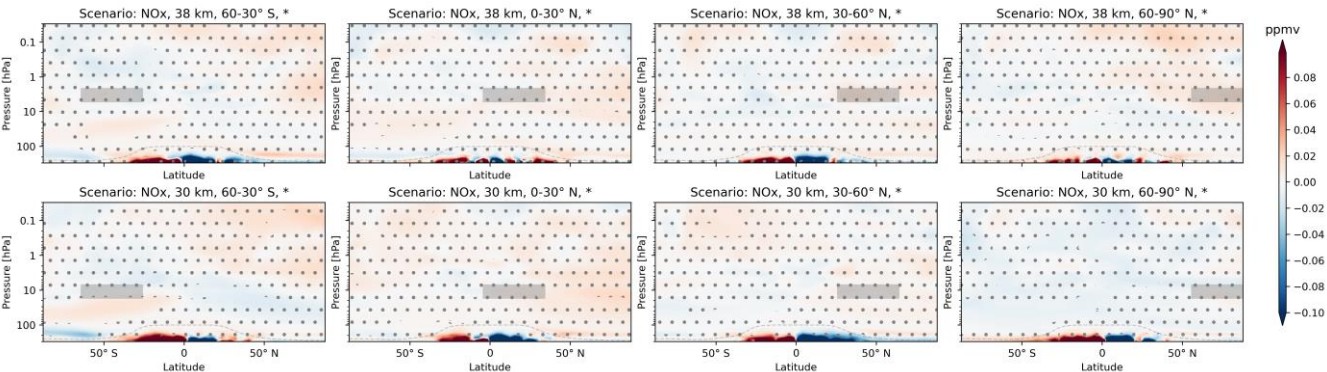

**Figure 21: Zonal mean of H$_2$O perturbation (volume mixing ratio, ppbv) for scenarios where NO$_x$ is emitted. The first and second row refer to 38 km and 30 km altitude of emission, respectively. Columns represent the latitude region of emission. The emission regions are shaded in gray. Dots indicate statistical insignificant results and the probability level is written in the title of each subplot.**

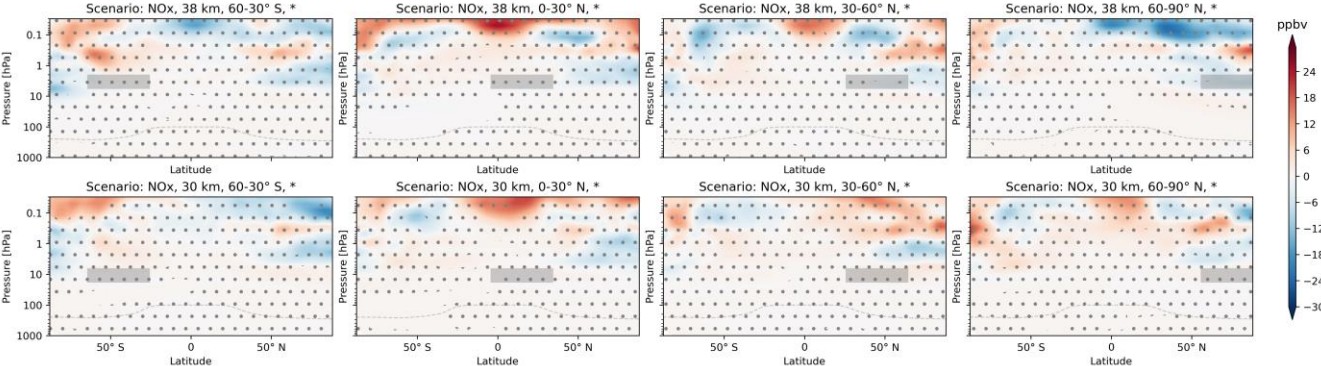

**Figure 22: Zonal mean of H$_2$ perturbation (volume mixing ratio, ppbv) for scenarios where NO$_x$ is emitted. The first and second row refer to 38 km and 30 km altitude of emission, respectively. Columns represent the latitude region**



of emission. The emission regions are shaded in gray. Dots indicate statistical insignificant results and the probability level is written in the title of each subplot.

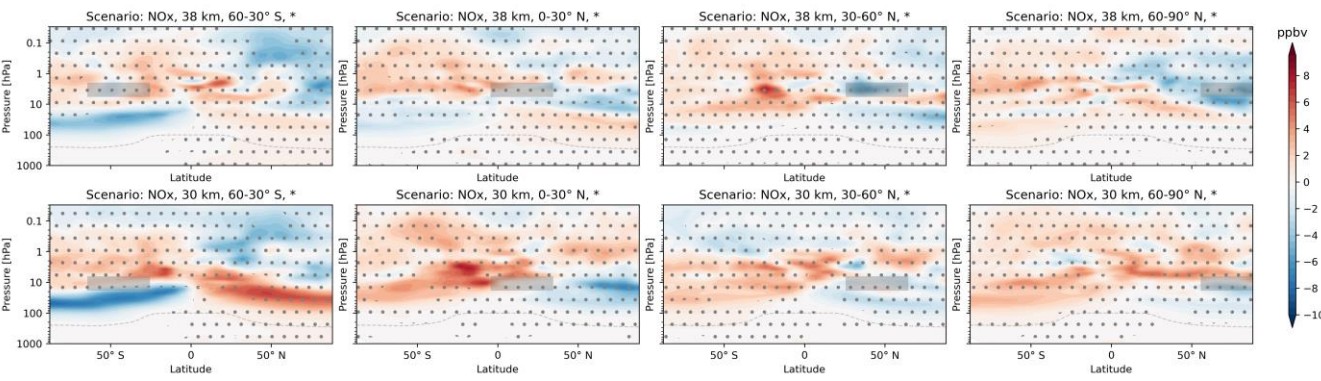

**Figure 23: Zonal mean of CH$_4$ perturbation (volume mixing ratio, ppbv) for scenarios where NO$_x$ is emitted. The first and second row refer to 38 km and 30 km altitude of emission, respectively. Columns represent the latitude region of emission. The emission regions are shaded in gray. Dots indicate statistical insignificant results and the probability level is written in the title of each subplot.**

### 11.3.4. Hydrogen emission

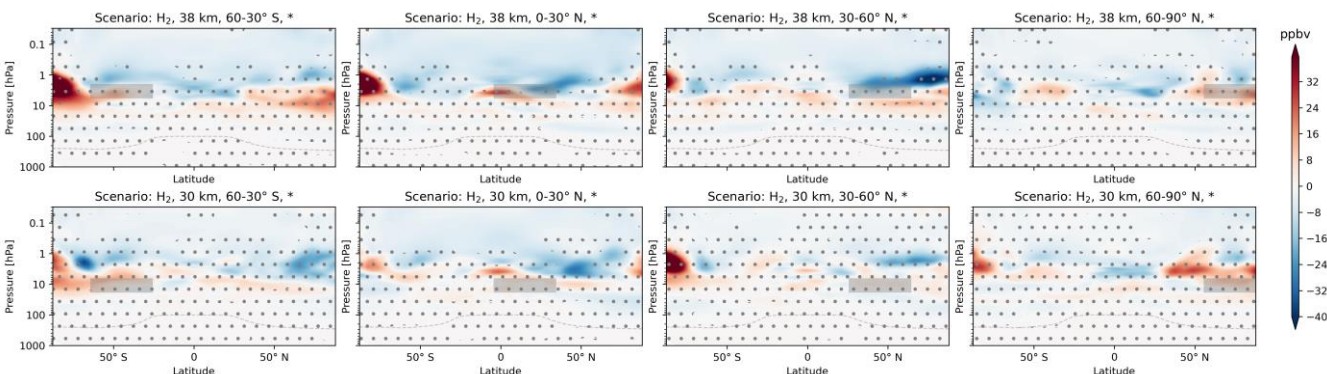

**Figure 24: Zonal mean of O$_3$ perturbation (volume mixing ratio, ppbv) for scenarios where H$_2$ is emitted. The first and second row refer to 38 km and 30 km altitude of emission, respectively. Columns represent the latitude region of emission. The emission regions are shaded in gray. Dots indicate statistical insignificant results and the probability level is written in the title of each subplot.**



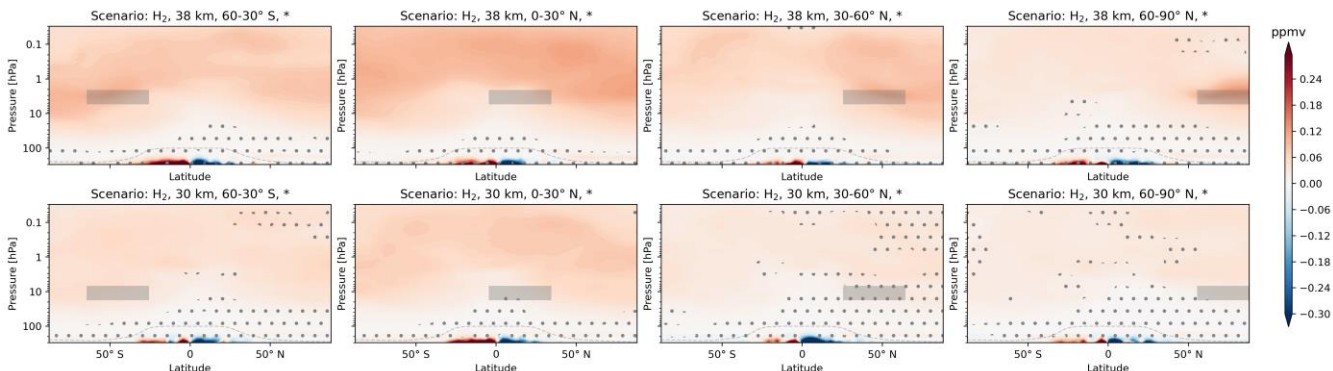


**Figure 25: Zonal mean of H₂O perturbation (volume mixing ratio, ppmv) for scenarios where H₂ is emitted. The first and second row refer to 38 km and 30 km altitude of emission, respectively. Columns represent the latitude region of emission. The emission regions are shaded in gray. Dots indicate statistical insignificant results and the probability level is written in the title of each subplot.**

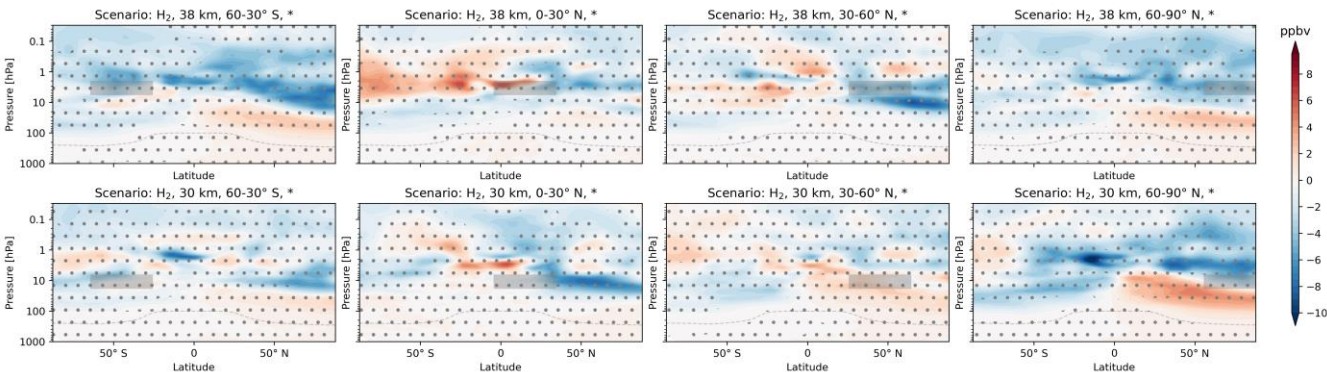


**Figure 26: Zonal mean of CH₄ perturbation (volume mixing ratio, ppbv) for scenarios where H₂ is emitted. The first and second row refer to 38 km and 30 km altitude of emission, respectively. Columns represent the latitude region of emission. The emission regions are shaded in gray. Dots indicate statistical insignificant results and the probability level is written in the title of each subplot.**





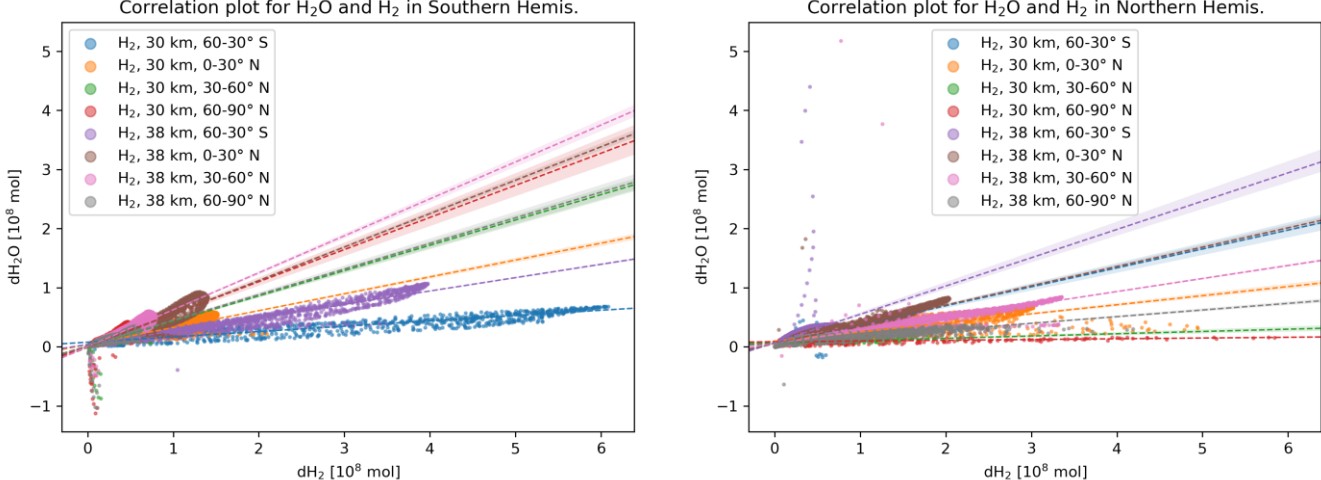


**Figure 27: Correlation of molar water vapour and hydrogen change for the Southern (left) and Northern (right) Hemisphere. The values are limited to statistically significant values with a 95 % confidence.**

### 11.3.5. Overview on atmospheric composition

**Table 5: Average atmospheric composition changes per molecule of emitted species**

| Emission | $\Delta H_2O$ | $\Delta H_2O$ / molecule | $\Delta O_3$ | $\Delta O_3$/ molecule |
|---|---|---|---|---|
| $H_2O$ | ~ 100 Tg | 85 Tg (Tmol/year)$^{-1}$ | -0.04 % | -0.03 % (Tmol/year)$^{-1}$ |
| $NO_x$ | ~ 0 Tg | 0 Tg (Tmol/year)$^{-1}$ | -0.15 % | -242 % (Tmol/year)$^{-1}$ |
| $H_2$ | ~ 5 Tg | 42 Tg (Tmol/year)$^{-1}$ | -0.02 % | -0.17 % (Tmol/year)$^{-1}$ |

| Emission | Sphere | $\Delta CH4$ [%] | $\Delta HO2$ [%] | $\Delta OH$ [%] | $\Delta CH4$/Tmol [%] | $\Delta HO2$/Tmol [%] | $\Delta OH$/Tmol [%] |
|---|---|---|---|---|---|---|---|
| $H_2O$ | Troposphere | -0,0014 | -1,12 | -0,15 | -0,001 | -0,9 | -0,1 |
| $H_2O$ | Middle Atmosphere | -0,0851 | 4,31 | 5,31 | -0,072 | 3,7 | 4,5 |
| $NO_x$ | Troposphere | -0,0015 | 0,05 | 0,07 | -2,218 | 71,0 | 107,2 |
| $NO_x$ | Middle Atmosphere | -0,0057 | -0,44 | 0,09 | -8,456 | -655,1 | 126,6 |
| $H_2$ | Troposphere | -0,0002 | 0,02 | -0,02 | -0,002 | 0,1 | -0,2 |
| $H_2$ | Middle Atmosphere | -0,0140 | 0,39 | 0,47 | -0,120 | 3,3 | 4,0 |


**Table 6: Relative atmospheric composition change of CH₄, HO₂ and OH (average) for the troposphere and middle atmosphere**





## 11.4. Radiative forcing

### 11.4.1. Water vapour radiative forcing

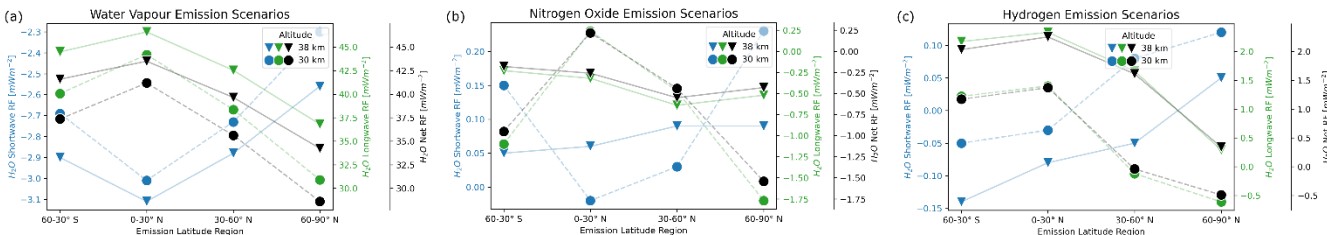


**Figure 28: The shortwave, longwave and net radiative forcing in blue, green and black, respectively, for water vapour (a), nitrogen oxide (b) and hydrogen (c) emission scenarios due to water vapour changes. For different altitude emission scenarios refer to markers. Latitude regions of emission are aligned on the x-axis.**

### 11.4.2. Ozone radiative forcing

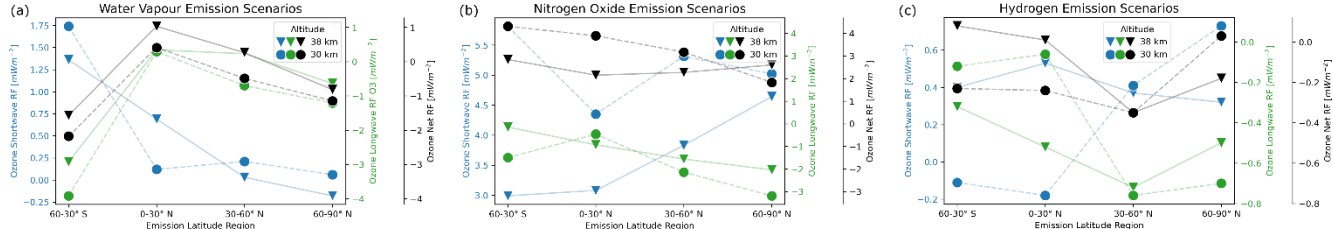


**Figure 29: The shortwave, longwave and net radiative forcing in blue, green and black, respectively, for water vapour (a), nitrogen oxide (b) and hydrogen (c) emission scenarios due to ozone changes. For different altitude emission scenarios refer to markers. Latitude regions of emission are aligned on the x-axis.**

### 11.4.3. Methane radiative forcing

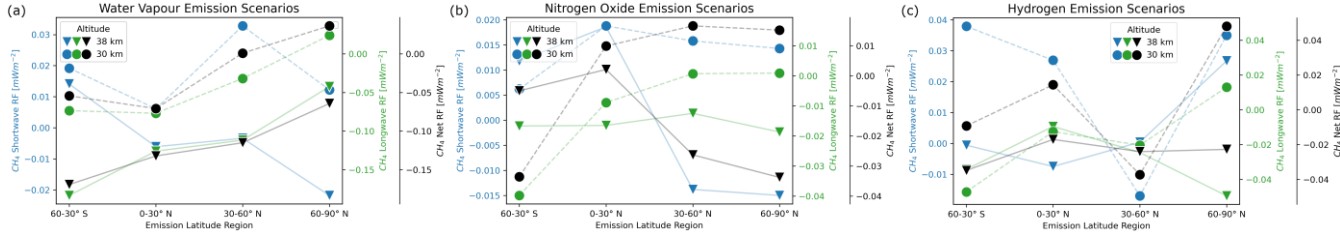


**Figure 30: The shortwave, longwave and net radiative forcing in blue, green and black, respectively, for water vapour (a), nitrogen oxide (b) and hydrogen (c) emission scenarios due to methane changes. For different altitude emission scenarios refer to markers. Latitude regions of emission are aligned on the x-axis.**




## 11.4.4. Individual radiative forcing and the related perturbations

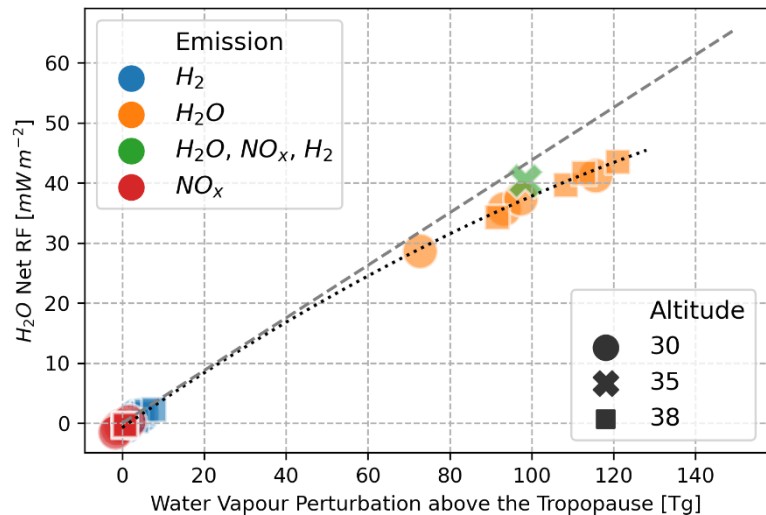


**Figure 31: Overview of H₂O net radiative forcing for all 24 simulations. Colours refer to the initial emission (H₂, H₂O and NOₓ) and markers to altitude of emission. The green cross for 35 km altitude refers to the publication of Pletzer et al. (2022) and is based on a global emission scenario (aircraft LAPCAT MR2) where H₂, H₂O and NOₓ are emitted simultaneously. The gray line refers to a correlation of RF and H₂O perturbation from another publication**
**(Fig. 8, Grewe et al., 2014a). The dotted black line is a two-dimensional polynomial fit with $ax^2 + bx + c$, where a = -0.88, b = 473.73 and c = -746.76 $Wm^{-2}Tg^{-1}$.**

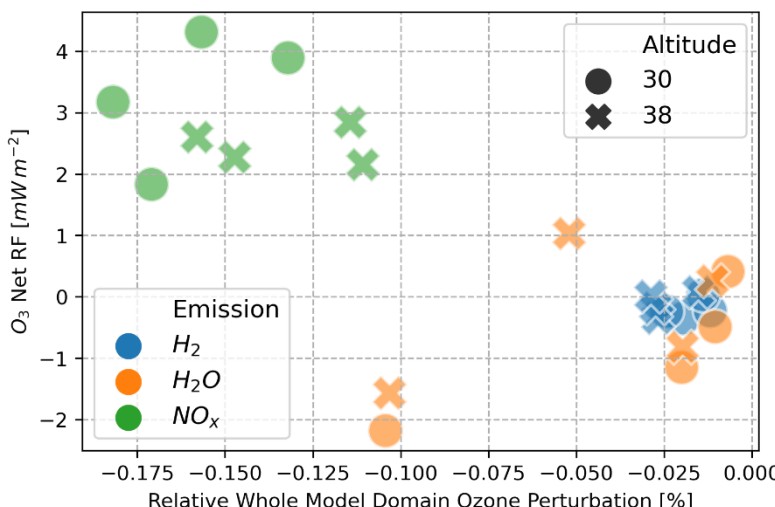

**Figure 32: Overview of O₃ net radiative forcing for all 24 simulations. Colors refer to the initial emission (H₂, H₂O and NOₓ) and markers to altitude of emission.**





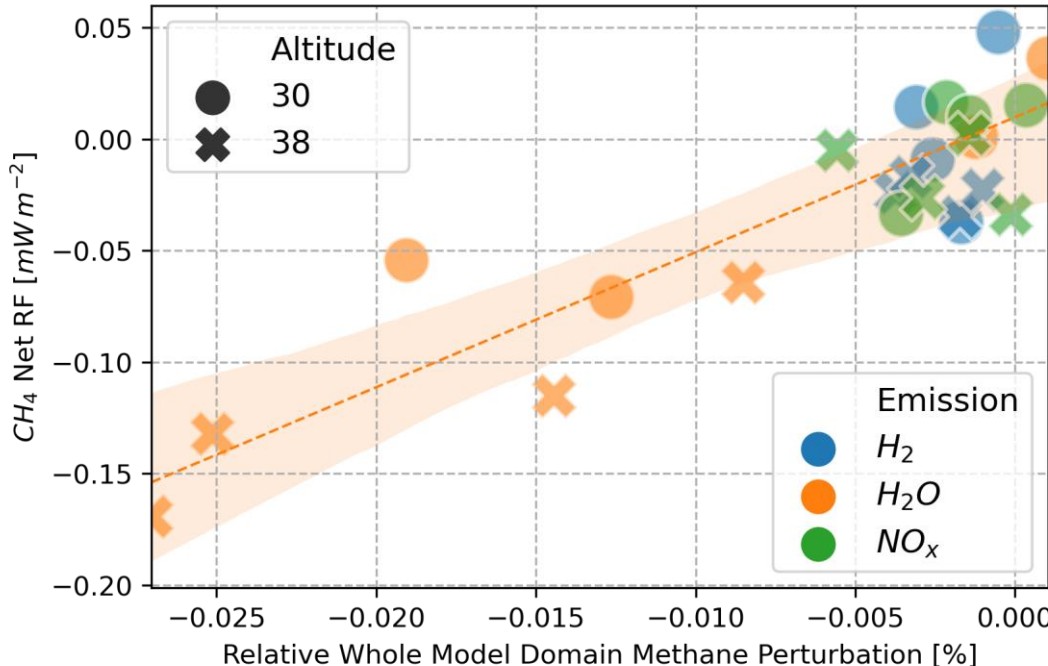


**Figure 33: Overview of CH₄ net radiative forcing for all 24 simulations. Colors refer to the initial emission (H₂, H₂O and NOₓ) and markers to altitude of emission. The regression line including the shaded areas depict the mean and standard deviation of the eight H₂O emission scenarios.**





## 11.5. Discussion of atmospheric composition changes and radiative forcing

**11.5.1. Water vapour atmospheric and radiative sensitivities**

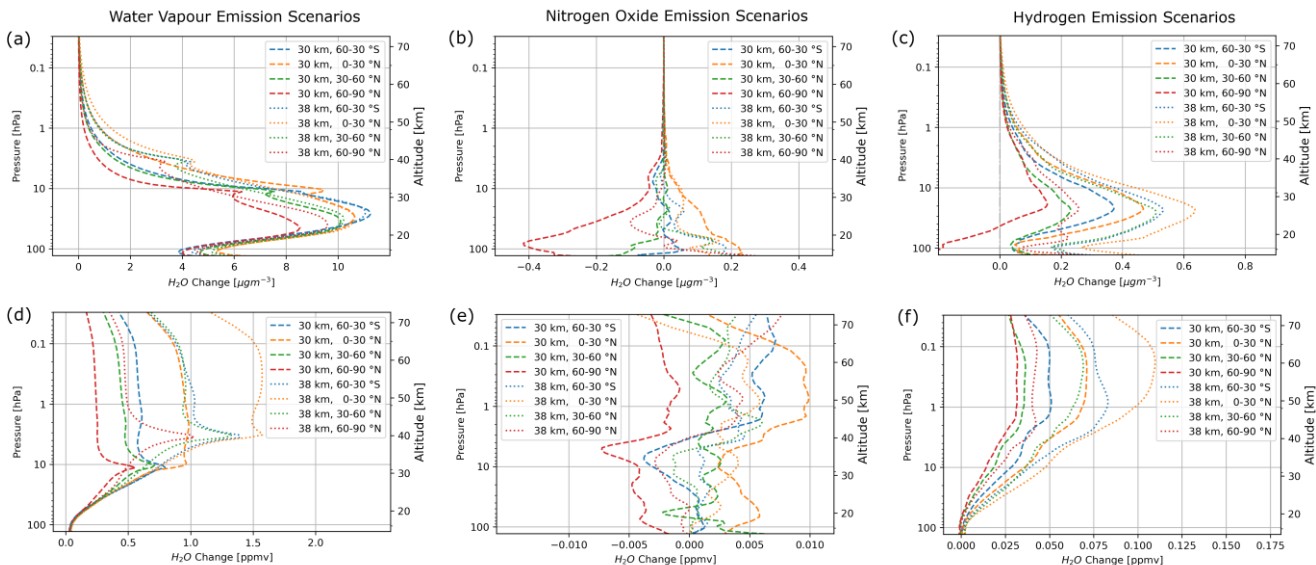

**Figure 34: Vertical distribution of global H₂O change as concentration (a, b, c) and volume mixing ratio (d, e, f). Subfigures (a, d), (b, e) and (c, f) depict H₂O change due to H₂O, NOₓ and H₂ emission respectively,**





## 11.5.2. Ozone atmospheric and radiative sensitivities

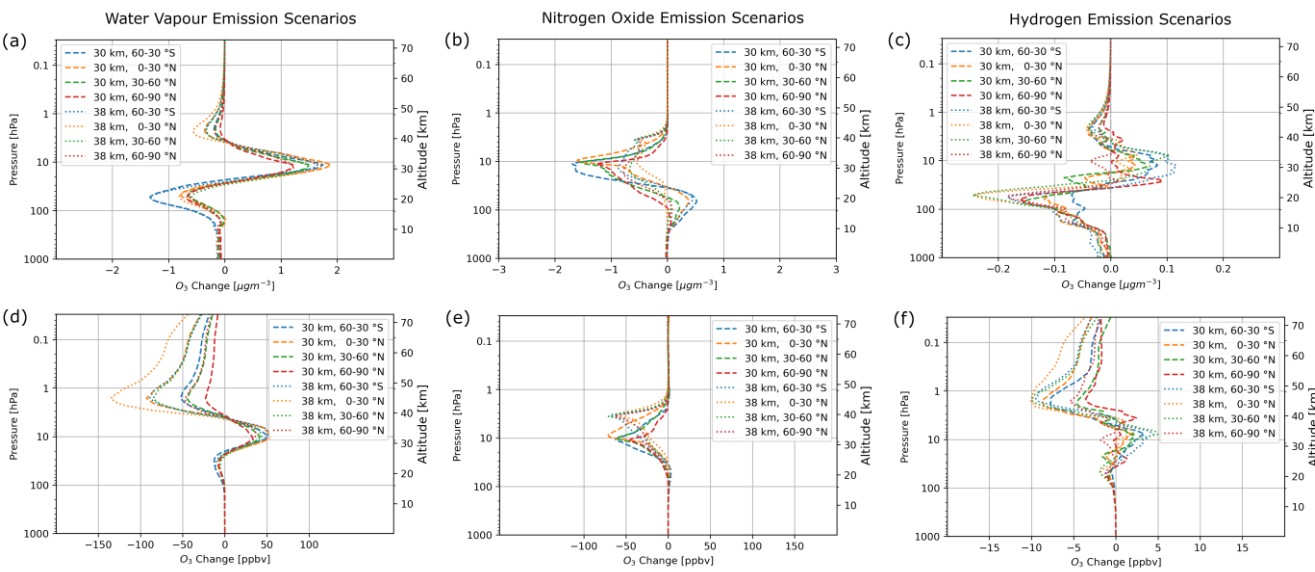


**Figure 35: Vertical distribution of global O₃ change as concentration (a, b, c) and volume mixing ratio (d, e, f). Subfigures (a, d), (b, e) and (c, f) depict O₃ change due to H₂O, NOₓ and H₂ emission respectively,**





## 11.6. Conclusion and discussion

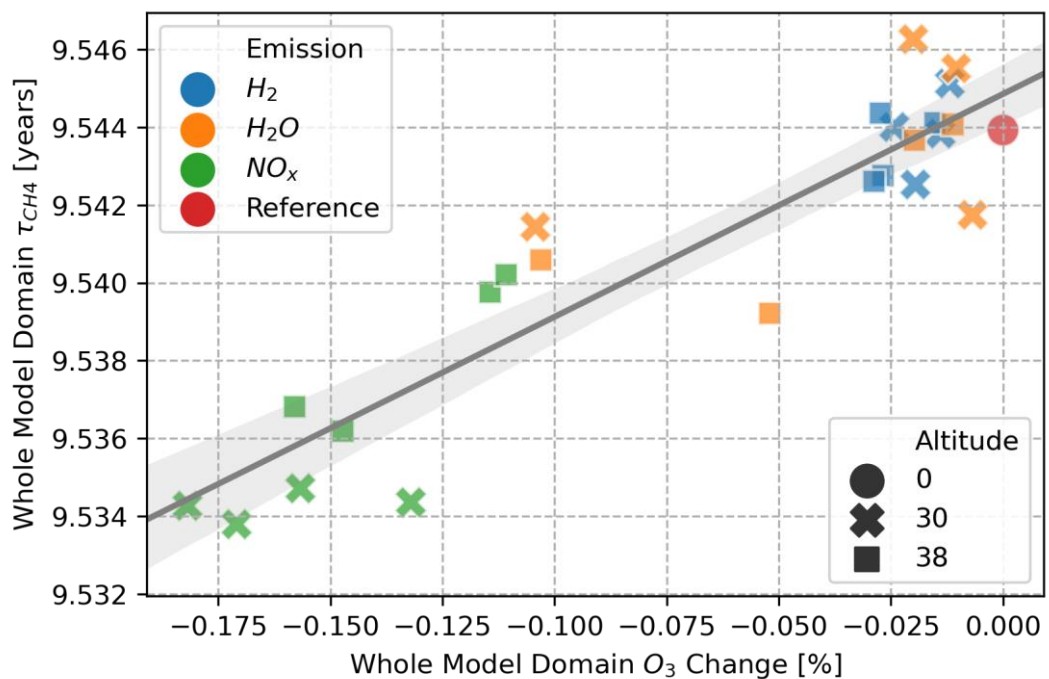

**Figure 36: Methane lifetime in relation to ozone change (both whole model domain) for all emission scenarios (legend). The gray regression line including the shaded area depict the 95 % confidence interval of all scenarios.**

## 12. Author contribution

Johannes Pletzer set up the EMAC model simulations, analyzed and post-processed the EMAC model results. Volker Grewe was involved in the discussion of the results and supported the writing of the document. The manuscript was written by
Johannes Pletzer, then reviewed by the authors and approved by all the authors.

## 13. Competing interests

The authors declare that they have no competing interests.



## 14. Acknowledgements

The model simulations have been performed at the German Climate Computing Centre (DKRZ). The resources for the
simulations were offered by the Bundesministerium für Bildung und Forschung (BMBF). We both thank kindly for this
opportunity.

The H2020 STRATOFLY Project and the H2020 MORE&LESS project have received funding from the European Union's
Horizon 2020 research and innovation program under Grant Agreement No. 769246 and 101006856, respectively.

We want to acknowledge the contribution by Davis et al. (2016) that enabled us to use satellite measurements for validation
of $H_2O$ and $O_3$ mixing ratios of EMAC results.

The main part of post-processing, data analysis and plotting has been done using the module xarray (version 0.20.1 Hoyer
and Hamman, 2017) and matplotlib.

For interpolating hybrid model level to pressure level we used the *interp_hybrid_to_pressure* function from geocat.comp
(Visualization & Analysis Systems Technologies, 2021).

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
