# Peer review of "Sensitivities of atmospheric composition and climate to altitude and latitude of hypersonic aircraft emissions"

_EGUsphere, 2023_

## Author Comment (AC1)

**Authors comment (AC1) to Referees comment (RC1)**

*Dear Referee,*

*on behalf of all authors I would like to thank you very much for taking your time to comment during the open discussion of the preprint. We very much appreciate your valuable propositions and included nearly all of them in our text. Our answers are in italic and blue.*

*Best regards,*

*Johannes Pletzer*

**General Comments**:

This paper addresses the potential impacts of a future fleet (~2050) of hypersonic aircraft emissions ($H_2O$, $H_2$, $CH_4$, and NOx) on middle atmosphere composition (i.e., $H_2O$, NOy, HOy, etc…) and the resultant chemical impacts on ozone. The stratospheric-adjusted radiative forcing from hypersonic aircraft perturbations is examined in "exhausting" detail. This paper will be a benchmark study for how perturbations from aircraft or any other vehicle emitting gases in the middle atmosphere affect climate forcing.

Overall, I find this work suitable (as is) for publication in ACP. I have a few comments below that the authors may find useful.

**Specific Comments:**

**Line 13:** "… can have a large

*We prefer to remain with "… can have a larger …" due to the comparison.*

**Line 20:** "… simulated years by one-third and thus cost and climate impact." Do you mean by running simulations in a shorter period, you are not contributing as much to global warming? This may be correct, but is that really necessary to point out (at least at the abstract level)?

*We removed this part from the text. Thank you for your viewpoint.*

**Line 26:** "…strong noise"? maybe "…"loud noise"?

*We changed "strong" to "loud".*

References seem to be duplicated, e.g., Jockel, et al., 2016a and Jockel et al. 2016b are the same paper. There are many examples of this type of duplication. Please double check all your references.

*We checked all references and removed duplicates. Thank you for pointing that out.*

**Page 2.** It would be nice for the "uninformed" reader what the primary difference is between a proposed supersonic and hypersonic aircraft are (i.e., relative to the speed of sound).

*Following your comment, we added this explanation: "Technically there are three categories of aircraft: subsonic aircraft that fly slower than the speed of sound, supersonic aircraft, whose speed exceeds the speed of sound, whereas the speed of hypersonic aircraft is at least five times the speed of sound."*

**Line 41:** Grammar, please reword. "One of the newest analyses the climate impact and the growth potential using projections of different technological development scenarios (Grew et al., 2021).

*We restructured the sentence for a better understanding. It now reads:*

*"One of the newest estimates analyses both, the climate impact and the growth potential, using projections of different technological development scenarios (Grewe et al., 2021)."*

**Line 51:** Not sure what you mean by "…climate change is manifold that of subsonic aircraft…"?

*We changed it to "… climate impact per revenue passenger kilometer "*

**Line 68:** (EMAC)j?  Is "j" a typo?

*It is a typo and was changed accordingly.*

**Line 72:** "In section three, we present the EMAC model…" You actually present this in sections 2.1. I believe you need to rename all the section numbers here.

*Thank you for pointing that out. The references were changed accordingly.*

**Line 88:** My next statement is a personal preference… The details of the EMAC model setup could easily be in an appendix. I'm not sure why the reader needs to muddle through all the sub-model process names. Maybe just have a more top-level model description in the main paper. One could probably reference a few EMAC publications in this section and only highlight what was important for this study.

*We very much appreciate your viewpoint. The extensive submodel descriptions are moved to the appendix (section 11.1).*

**Section 2.3**. I would highly recommend a table that discusses the model configurations. You do state in words what simulations are performed in this work – however, I had to read sections 2.1 through 2.3 several times to figure out exactly what you did.  A table would help the reader.

*We added a table to subsection 2.2 (Table 1: Overview of key properties used in the model setup). Enumeration of other tables was changed accordingly.*

**Specified Dynamics**. Based on the discussion in section 2, it seems you are using "observed" specified dynamics met fields (present-day ERA-Interm) for the 2050 aircraft scenarios (line

129). You should probably say in a few words why this is justifiable, instead, using model specified dynamics fields from an interactive climate run for the year 2050.

*We extended the text on specified dynamics with the following reference:*

*"Note that the impact of a 2050 meteorology would be – coarsely estimated – a 8-10 % strengthening of the middle atmospheric circulation, which would reduce atmospheric perturbations from hypersonic aircraft accordingly (subsubsection 7.1.2, Pletzer et al., 2022)."*

**Section 2.4.** Enhancing the efficient use of computer time. Very interesting approach!! This could be adopted by many research studies that run a model to a steady state condition.

*Thank you very much for the feedback. The approach is currently being applied for similar simulations inhouse and might become even more useful for higher resolution simulations.*

**Section 3.2.** Can you put the emissions ($H_2O$, $H_2$, NOx) in context, i.e., how many aircraft are considered in this scenario? You mention the LAPCAT-PREPHA aircraft design; but are the number of planes significant to other published SST, HSCT scenarios. This is important when you discuss later the impact of emissions on Ozone and $CH_4$ lifetime.

*The scenario considers on average 206 flights per day, which is nearly equivalent to numbers from e.g. Ingenito (2018):*

https://www.sciencedirect.com/science/article/pii/S0360319918331379

*We extended the sentence describing LAPCAT-PREPHA in section 3.2 with this piece of information.*

**Line 260.** "For each emitted trace gas ($H_2O$, NOx, $H_2$) we have a total of eight simulations, which sum up to 24 simulations in total. The annual magnitude of emitted trace gases is 21.24 teragram of $H_2O$, 0.031 teragram $NO_2$ of NOxand 0.236 teragram of $H_2$." Question: the scenarios are designed to emit one emission species per simulation. I understand why you chose to emit on species per scenario. However, you might want to mention why did you not combine all the emissions of $H_2O$, $NO_2$, and $H_2$ in one simulation to examine how the chemistry responded (e.g., ozone chemistry)? I do note that later in the paper you did discuss this topic in section 7.3 and 7.4. Possibly point to this section for later discussion.

*We added a reference in line 260 to the discussion in section 7.4.*

**Figure 3.** The color bar goes from -6.0 to >4.8. Question. Why are showing negative colors in your color bar? Seems to me that when you inject $H_2O$ you will always have a positive perturbation?

*We changed the colorbar accordingly.*

**Figure 4; section 4.1.2.1 Ozone.** Why is ozone increasing in region near 10hPa? I.e., is this HOx chemistry interacting with NOx chemistry?

*If compared to Figure 18 (appendix) NOx changes seem to be inversely correlated with O3 between 100-10 hPa. Additionally, HNO3 is increased at 10hPa. The increased oxidation*

*capacity is most probably the cause of these changes. Here is a Figure with HNO3 changes for reference:*

[Figure]

**Figure 5: section 4.1.2.3 Methane.** I understand why CH₄ is decreased with H₂O emissions – why does it increase? What is the mechanism?

*That is a good question that we cannot answer with much certainty. Coarsely described at 100-10 hPa there is, first, increase/decrease of CH4 divided among Hemispheres and, second, changes appear rather at polar regions than in the tropics. We can report that between 100-10 hPa CH4 changes overlap with changes of C2H6 (not shown). Note that we do not want to imply a causal relation here. Additionally, we see an overall reduction of the HNO3 + OH reaction rate, where CH4 increases appear.*

*Other than that, CH4 increase appears in areas, where Cl and HCl concentration is lower than in the reference scenario. We are referring to the reaction Cl + CH4 → HCl + CH3O2, which we did not track with additional diagnostics unfortunately.*

*To conclude briefly, the extratropical regions, where most of the changes appear, is very much influenced by transport and it is generally difficult without a model setup tailored to track methane changes to find correct answers. We do hope that we could shed some light on this topic. We would prefer not to include this particular information in the publication, since in our opinion it is beyond the scope of this work.*

**Section 4.4.1.** It seems that the perturbations (emissions) are not impacting the CH₄ lifetime in any significant way!

*As shown in Table 4 (formerly Table 3) absolute CH4 lifetime changes are mostly significant for nitrogen oxide emission scenarios. There, tropospheric oxidation chemistry is more active. In our opinion, in the middle atmosphere areas of CH4 increase and areas of CH4 decrease might nearly cancel each other in terms of methane lifetime and the impact of the tropospheric domain is therefore more dominant (for nitrogen oxide emission scenarios).*

**Section 4.4.3.1.** This statement is interesting. "The order of magnitude of changes per molecule of emitted species shows that a molecule of $H_2$ is roughly 50 % as effective in enhancing the mid-atmospheric $H_2O$ concentration as a molecule of emitted $H_2O$ (Tbl. 3)." Can you discuss the chemistry here? I.e., $H_2$ +OH => $H_2O$ + H. $H_2$ can also be converted to H and OH (i.e., HOx). The HOx can also be converted back to $H_2O$.

*We added an explanation:*

*The relevant reactions include both, loss and production of H2O. Chemically, the production is initiated directly by reaction H2 + OH → H2O + H and indirectly by HO2 + OH → H2O + O2. The latter is facilitated by the general increase of HOx and included in a H2O-HOx cycle via reaction H2O + O(1D) = 2 OH. Increased oxidation, e.g. methane and nitric acid oxidation, contributes a small amount as well. The net production of H2O briefly discussed in subsection 7.2.*

**Section 4.4.3.2.** This section and the previous are very informative on the reactivity of emitted H2! Very nice. "Interestingly, while in absolute values the $H_2$ emissions are of minor importance to the $O_3$ depletion, the average effectiveness in destroying $O_3$ is roughly 5-6 times larger for $H_2$ than for $H_2O$ (Tbl. 3)." I.e., $H_2$ (vs $H_2O$) has opposite effect on Ozone (compared to $H_2O$; section 4.4.3.1). I do note that this is partially discussed in section 7.2.

*Thank you very much for your feedback. We appreciate it very much.*

**Figure 10 is a very nice summary figure.**

*Thank you very much for your kind words.*

---

## Author Comment (AC2)

**Authors comment (AC2) to Referees comment (RC2)**

*Dear Referee,*

*on behalf of all authors I would like to thank you very much for taking your time to comment during the open discussion of the preprint. We addressed all of your comments and used the propositions to improve the manuscript for the reader, wherever possible. Please find our answers below in italic blue just to your comments (black).*

*Best regards,*

*Johannes Pletzer*

This paper presents an assessment of the impacts of H2, H2O and NOx emissions from high-flying hypersonic aircraft. This is based on a series of calculations with a detailed 3D chemistry-climate model which are analysed for the changes in composition and radiative forcing.

This is an interesting topic and well within the scope for ACP. The tools and methodology are entirely appropriate for the problem and there are a number of useful results in this paper. However, in my opinion the paper is not suitable for publication in its current form. The paper is very long and I find the presentation very chaotic. I think the main points could be communicated in much less text and far fewer figures. There are also a number of typos and mistakes in the text which also detract from the overall impression.

Therefore, I think the paper needs major revisions. My main comments are below. Given that my advice is a significant shortening of the paper I have not provided comments and all parts of the text.

*We appreciate your support of our methodology and results and would like to thank you for pointing at typos in the text.*

*The most stressed comment refers to the length of the paper.*

*We would like to ask the Editor, Prof. John Plane, to give his thoughts on this matter. From our perspective, we received the feedback to publish as is from RC1 especially pointing out the broad coverage of results and in turn, we were asked by RC2 to apply major revisions and basically shortening. In this respect, the two reviews contradict each other. We are happy to bundle certain parts of the main text as a supplement if it helps the reader. In our opinion, parts of subsection 2.3 (SWOOSH comparison and parts of section 5 (Radiative Forcing) could potentially be moved to the appendix or bundled as a supplement. While subsection 2.4 (speed-up technique) could also be moved to the appendix or a supplement, we want to refer to RC1, which found the approach very interesting and relevant for other studies and we would very much like to keep it in the main text. The appendix as it is now could also be bundled as a supplement. There are multiple options and we would require guidance from the editor.*

Abstract.

I think that the abstract should be more quantitative. It should state that the results are based on 3D CCM simulations. I don't understand the message in the final sentence.

*We included a remark, which kind of model we used for obtaining the results. Thank you for pointing that out.*

*Note that the final paragraph of the abstract was revised as a result of the first referee's comments. We think that the final sentence has become clearer.*

Introduction.

I cannot see where you define/explain what a hypersonic aircraft is. You need to justify your use of 30 and 38 km for the emissions.

*We added a technical explanation of subsonic, supersonic and hypersonic aircraft, as suggested by both, you and the first referee:*

> *"Technically there are three categories of aircraft: subsonic aircraft that fly slower than the speed of sound, supersonic aircraft, whose speed exceeds the speed of sound, whereas the speed of hypersonic aircraft is at least five times the speed of sound."*

The section numbering in the final paragraph is wrong.

*The section numbering has been corrected. Thank you for pointing that out.*

Model Experiments

The model experiments should be given labels, including the new runs for this paper. Sometimes they are referred to as 'ours' which is not clear. The text says that all runs use nudged dynamics? Is that the case even for the future composition? The simulation years are also always in the range 2007-2017, or so. This is the case for the future scenarios, yes? Overall there are a number of things to do to make the model simulations clear for the reader.

*Appearances of "ours" in the context of the model setup were replaced with the label "HS-sens", which is now introduced in subsection 2.1.*

*We can confirm that all runs use specified dynamics (nudged towards ECMWF data). In contrast, the chemical lower boundary conditions are based on 2050 (see new Table 1). We partly removed time ranges (e.g. "2013-2016") throughout the preprint, e.g. from figure captions, to avoid misunderstandings. The calculation of results is further clarified in section 4, first paragraph, with the following sentence:*

> *"Note that all presented data in this section is based on multi-annual mean model results with both, specified meteorology (2013-2016) and 2050 source gas emissions."*

*The subsection 2.1 was revised and shortened. Subsection 2.2 was further divided in subsubsections to structure the method section more clearly. A brief extension was added the the subsection "2.2 EMAC model setup":*

> *"The setup combines boundary conditions of 2050 surface emissions and nudging to present day meteorology."*

*The quick-look table (Table 1), which was added as a result to RC1, was extended by two lines to include information on meteorology and surface emissions.*

Satellite Validation

I don't see how comparing present-day observations with model runs which use future composition adds anything beyond the comparison of the same model with realistic present-day composition. If the only difference in the model is the source gas loadings then one has to say that the present-day model has been evaluated and you have just changed the boundary conditions. If you want to show the impact of the source gas changes in the model then that is a different issue.

*The approach you suggest is certainly another viable option.*

SWOOSH may be a long climatology but only 4 years are used (2013-2016). What are the main datasets in that make up SWOOSH in this period? A 40% or so difference in stratospheric water vapour seems very large to me. How does CH4 compare? What about total hydrogen (2CH4 + H2O (+H2?)) Is there an issue with the age of air (too much CH4, too little H2O) at a certain location. I don't think a 40% error in H2O can be ignored.

*Information on SWOOSH data is presented in subsubsection 2.3.1. It is introduced that for the years 2013-2016 SWOOSH dataset consists of Aura MLS data. Additionally, a comparison to other instruments is included. In our opinion the information on SWOOSH is sufficient and does not require any extention. However, we are happy to add further information*

*While we find all the questions very interesting, we focused on the two most important climate drivers, $O_3$ and $H_2O$. In our opinion, everything beyond would require specific publication focused only on model validation. Please note that EMAC results show a systematic cold bias, and hydrogen or methane oxidation should not be the issue (Jöckel et al, 2016; Pletzer et al, 2022).*

*The referee is correct. 40 % sounds like a lot. However, previous publications show that model results agree well on water vapour perturbations in both, perturbation patterns and total magnitude (Pletzer et al, 2022; Kinnison et al, 2020). Following the comparison to satellite data, our viewpoint is that background water vapour is less important for the magnitude of middle atmospheric water vapour perturbations than compared to e.g. transport time to the troposphere. The latter was quantified in Fig. 8. It would be interesting to see similar results from other models.*

Figures

There are too many figures (35!). I know that some are in an Appendix but they are referred to throughout the text as though they are main figures and not supplementary ones. In effect, the reader is reading a paper of 35 figures which is way too long. Some panels in the figures are small with small font size.

*The panels in the figures were increased in size. This includes zonal mean figures (Figs. 3-7, 17, 18, 20-26) and figures on radiative forcing (Figs. 9, 28-30).*

Tables

The tables need checking and tidying up. Table 2 is just explaining a legend which appears in later tables without any reference back to Table 2. It would be simpler just to put this code in the heading of e.g. Table 3.  The second 'Magnitude' column of Table 1 must also be 'per year'? Why is there the need for the final three columns of Table 1 – they are all the same.

*We extended the 'Magnitude' column of Table 2 (formerly Table 1) with '/year'. Thank you for pointing that out. We removed the final three columns.*

*The information in Table 3 (formerly Table 2) is also used in multiple figures and is therefore needed not only for Tables 4 and 5 (formerly Table 3 and 4). We added a reference to Table 3 in Table 4 and 5 (formerly Table 3 and 4).*

Results

Please shorten and rationalise the text and figures that are used to communicate the main results. The description of the the modelled changes for different latitudes/altitudes can be covered quite concisely but the relevant mechanisms at work should also be discussed (e.g. HOx, NOx chemistry etc).

*We very much agree with your viewpoint that the publication covers many aspects.*

*We would prefer to not focus the relevant mechanisms of HOx, NOx chemistry, since another publication already focuses very much on the chemistry (Kinnison et al, 2020).*

*Please see also discussion a the top of the reply concerning the length of the paper.*

**Literature references**

[1]

P. Jöckel *et al.*, 'Earth System Chemistry integrated Modelling (ESCiMo) with the Modular Earth Submodel System (MESSy) version 2.51', *Geosci. Model Dev.*, vol. 9, no. 3, pp. 1153–1200, Mar. 2016, doi: 10.5194/gmd-9-1153-2016.

[1]

D. Kinnison, G. P. Brasseur, S. L. Baughcum, J. Zhang, and D. Wuebbles, 'The Impact on the Ozone Layer of a Potential Fleet of Civil Hypersonic Aircraft', *Earth's Future*, vol. 8, no. 10, Oct. 2020, doi: 10.1029/2020EF001626.

[1]

J. Pletzer, D. Hauglustaine, Y. Cohen, P. Jöckel, and V. Grewe, 'The climate impact of hydrogen-powered hypersonic transport', *Atmospheric Chemistry and Physics*, vol. 22, no. 21, pp. 14323–14354, Nov. 2022, doi: 10.5194/acp-22-14323-2022.